# PERM: A PARAMETRIC REPRESENTATION FOR MULTI-STYLE 3D HAIR MODELING

**Chengan He**[1][*] **Xin Sun**[2], **Zhixin Shu**[2], **Fujun Luan**[2], **Sören Pirk**[3],
**Jorge Alejandro Amador Herrera**[4], **Dominik L. Michels**[4], **Tuanfeng Y. Wang**[2],
**Meng Zhang**[5], **Holly Rushmeier**[1], **Yi Zhou**[2]
[1]Yale University    [2]Adobe Research    [3]Kiel University    [4]KAUST    [5]NJUST
`{chengan.he,holly.rushmeier}@yale.edu`
`{xinsun,zshu,fluan,yangtwan,yizho}@adobe.com`
`sp@informatik.uni-kiel.de`
`{jorge.amadorherrera,dominik.michels}@kaust.edu.sa`
`mengzephyr@njust.edu.cn`

## ABSTRACT

We present PERM, a learned parametric representation of 3D human hair designed to facilitate various hair-related applications. Unlike previous work that jointly models the global hair structure and local curl patterns, we propose to disentangle them using a PCA-based strand representation in the frequency domain, thereby allowing more precise editing and output control. Specifically, we leverage our strand representation to fit and decompose hair geometry textures into low- to high-frequency hair structures, termed guide textures and residual textures, respectively. These decomposed textures are later parameterized with different generative models, emulating common stages in the hair grooming process. We conduct extensive experiments to validate the architecture design of PERM, and finally deploy the trained model as a generic prior to solve task-agnostic problems, further showcasing its flexibility and superiority in tasks such as single-view hair reconstruction, hairstyle editing, and hair-conditioned image generation. More details can be found on our project page: `https://cs.yale.edu/homes/che/projects/perm/`.

## 1 INTRODUCTION

3D hair modeling, as a crucial and expensive component in the realm of digital humans for industries like gaming, animation, VFX, and virtual reality, combines the complex and artistic processes to model the geometry of individual strands to create specific hairstyles in the 3D environment. With the recent availability of high-quality 3D hair data (Hu et al., 2015; Shen et al., 2023; Zhou et al., 2018), machine learning methods have emerged for automatic hairstyle synthesis (Zhou et al., 2023; Sklyarova et al., 2023b), and 3D hair reconstruction from images (Wu et al., 2022; Zheng et al., 2023; Takimoto et al., 2024; Kuang et al., 2022; Zhou et al., 2018) and monocular videos (Wu et al., 2024; Luo et al., 2024; Sklyarova et al., 2023a).

Despite their achievements, these methods often overlook the inherent *biscale* nature of hair, where the global structure defines the hair flow, volume and length and the local structure defines the strand's curl patterns. Neglecting these biscale variations not only limits the quality of hair generation and reconstruction (illustrated in the comparisons in Fig. 7 and Fig. 10), but also restricts the editing capability on different scales. Moreover, most of the models proposed in existing methods are heavy and *task-specific*, lacking the generalization capability to different down-stream tasks. In contrast, the field of human body modeling has well-accepted lightweight parametric models like SMPL (Loper et al., 2015) and MANO (Romero et al., 2017), which designed disentangled coefficients for pose and shape and are widely applied as a generic prior in deep learning applications such as human reconstruction and animation. To address this gap, we propose PERM, a paramet-

---

[*]The work was mainly conducted at Adobe Research.

ric representation of 3D human hair that is both lightweight and generic for various hair-related applications.

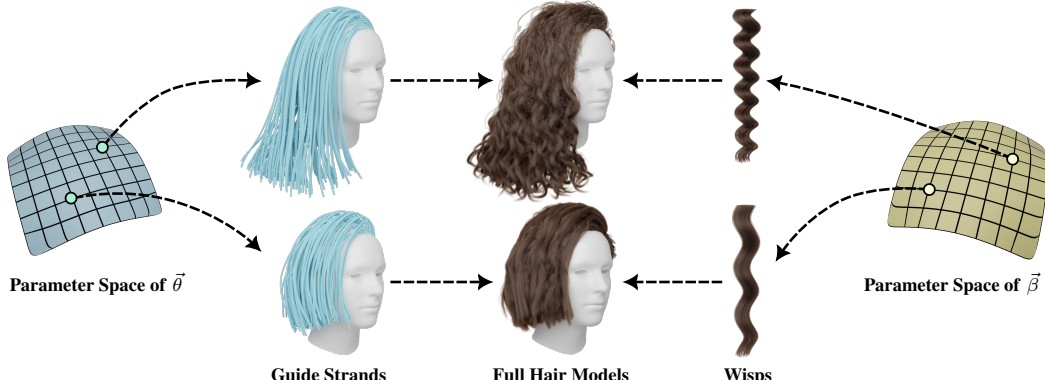

**Parameter Space of $\vec{\theta}$**     **Guide Strands**     **Full Hair Models**     **Wisps**     **Parameter Space of $\vec{\beta}$**

Figure 1: PERM is a learned parametric representation of 3D human hair that is designed with disentangled parameters $\vec{\theta}$ and $\vec{\beta}$ to respectively control the global hair structure (represented as guide strands) and local curl patterns (represented as wisps).

PERM is novelly formulated as $\mathcal{M}(\vec{\theta}, \vec{\beta}, \mathcal{R})$, where the output hair is conditioned on the guide strand parameter $\vec{\theta}$ for its global structure, the hair styling parameter $\vec{\beta}$ for its curl patterns, and a pre-defined root set $\mathcal{R}$ for the hair density (as illustrated in Fig. 1). To independently control the global structure and curl pattern, we propose a strand representation based on principal component analysis (PCA) in the frequency domain, which naturally aligns with the cyclical nature of hair growth (Hoover et al., 2023). We use around three millions of different strands to train the PCA model and compare it with several deep neural network-based models. Despite the PCA model's simplicity, it demonstrates high effectiveness in forming a compact subspace for strands that preserves geometry details much better than the deep neural network alternatives while demanding much less computation and memory consumption.

The learned principal components form an *interpretable* space for hair decomposition, where the initial components capture low-frequency information, such as the rough direction and length, while the subsequent components encode high-frequency details like curliness. As visualized in Fig. 2, this representation allows for the intuitive extraction of smooth guide strands from the given wisp, closer to what artists really use when grooming 3D hair in industrial software like Maya XGen (Autodesk, 2024).

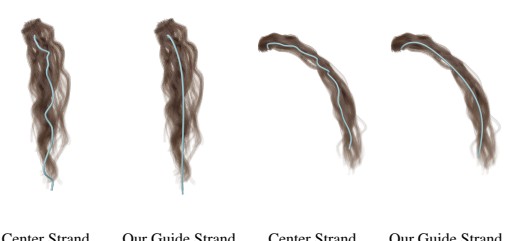

Center Strand     Our Guide Strand     Center Strand     Our Guide Strand

Figure 2: Guide strand of a wisp from PCA.

For parameterizing the full hair, we follow previous literature to compute the $uv$ of the scalp, and map the geometry features of each strand onto a 2D texture based on its root position on the scalp. With the strand PCA representation, we can fit and store the dominant coefficients into *guide textures* for guide strands and higher-order coefficients into *residual textures* for detailed strand patterns. These decomposed textures are then used to train a framework composed of multiple generative models as illustrated in Fig. 4.

After training PERM with $20k$ different hair styles, we obtain a compact, robust and editable generative hair model that can function as a generic hair prior for solving task-agnostic problems. We demonstrate its capability across multiple applications, including single-view hair reconstruction and hairstyle editing, such as changing a hairstyle from smooth to bouncy while maintaining a similar haircut. Despite not being trained specifically for any of these tasks, PERM achieves performance equivalent or superior to state-of-the-art task-specific alternatives in our experiments. Moreover, we introduce a novel application of using PERM-generated hair for conditional image generation and editing hairstyles in the 3D latent space. A demo of these applications is provided in Fig. 3.

*"wavy and short hair, white sweater"*

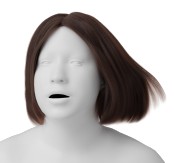 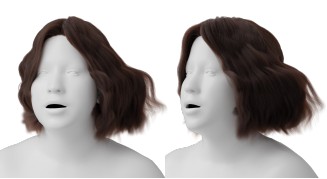 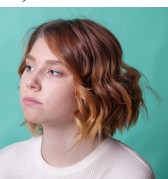

Input Image    Single-view Recon.    PERM Parameter Editing    Hair-conditioned Image Generation

Figure 3: An application demo of PERM, where we first fit PERM parameters to reconstruct 3D hair from the input image, then edit the parameters to change its style from straight to wavy, and finally use the edited hairstyle and certain viewing direction as a condition for image generation with Adobe Firefly (Adobe, 2024).

## 2   RELATED WORK

Due to the limited availability of 3D hair data, early hair modeling works (Wang et al., 2009) primarily focused on generating variations from a single hairstyle using methods like PCA. With the recent release of high-quality 3D hair datasets (Hu et al., 2015; Shen et al., 2023; Zhou et al., 2018), machine learning methods have emerged to tackle various hair-related tasks. Hair representations used in these methods are generally categorized into two types: volumetric hair representation and strand-based hair representation.

**Volumetric Hair Representation**   Considering the volumetric nature of hair, Saito et al. (2018) first proposed a volumetric hair representation that converts a 3D hair into a combination of a 3D occupancy field and an orientation field, where strands could later be generated from discretized voxels. As an efficient spatially-aware intermediate representation for network training, this representation has been widely adopted in subsequent works for a range of hair-related applications, spanning from sketch-based hair modeling (Shen et al., 2020), video-based dynamic hair modeling (Yang et al., 2019; Wang et al., 2023b), to 3D hair reconstruction from images (Zhang & Zheng, 2019; Takimoto et al., 2024; Kuang et al., 2022) or monocular videos (Wu et al., 2024; Sklyarova et al., 2023a). However, since discrete volumetric fields are not well-suited for representing curly and kinky hair, another line of research has sought to improve this volumetric representation. These advancements include implicit volumetric representations (Wu et al., 2022; Zheng et al., 2023) and hybrid representations that attach multiple neural volumetric primitives to pre-obtained sparse guide strands (Wang et al., 2022; 2023a).

**Strand-based Hair Representation**   Since real human hair consists of individual strands, the most intuitive way to represent it is as a collection of strands, where each strand can be represented as a 3D polyline with a fixed number of points. Zhou et al. (2018) adopted this strand-based representation and proposed to organize strand features as a 2D hair texture according to the scalp $uv$ parameterization, which facilitates the training of 2D convolutional networks. This idea has been expanded by subsequent works, incorporating more advanced networks such as U-Net (Ronneberger et al., 2015) and diffusion models (Ho et al., 2020) to solve tasks such as neural hair rendering (Rosu et al., 2022), text-conditioned hair generation (Sklyarova et al., 2023b) and 3D hair capture from monocular videos (Sklyarova et al., 2023a). While these diffusion-based methods produce impressive results, they typically involve a heavy denoising process and lack a compact latent space for hair reconstruction and editing. Recent work on 3D Gaussian Splatting (Kerbl et al., 2023) offers a potential enhancement to this representation, where Gaussian splats can be attached to each segment of the strand. This hybrid representation enables the joint learning of both hair geometry and appearance, showing promise in 3D hair capture (Zakharov et al., 2024; Luo et al., 2024; Zhou et al., 2024). However, all these strand-based works are task-specific and neglect the biscale nature of hair, thereby limiting the quality and versatility of the generated or reconstructed hairstyles.

**Generative Hair Model**   GroomGen (Zhou et al., 2023), most related to our work, learns a hair prior using hierarchical latent spaces. However, their hierarchy is only for the low-resolution hair texture, which refers to a sparse set of strands, and the high-resolution hair texture, which refers to the final full hair, is upsampled in a deterministic way. Although they denote the sparse strands

as guide strands, those are just like the center strands as illustrated in Fig. 2 without the filtered structure. They do not have the disentangled parameters for controlling the overall hair structure features and the detailed hair curl patterns, which leads to a different architecture design from our method.

While GroomGen reported promising results in their paper, we found it is both theoretically and experimentally unreliable through our experiments. Its VAE-based strand representation struggles to preserve the strand structure and the entire architecture is very unstable to train. We further found that some curly hairstyles showcased in the paper are unachievable with the described strand resolution. We include our experiments of GroomGen in Appendix C.4.

**PCA for Hair**  The biology community has widely adopted PCA to analyze and classify different hair types in the real world (Hoover et al., 2023; De La Mettrie et al., 2007; Loussouarn et al., 2007). We transfer these biological insights into the machine learning domain, introducing the first neural network architecture to incorporate a PCA-based strand representation for human hair. While Wang et al. (2009) previously explored PCA-based representations for variations of a single hair model, we aim at generating high variety of hairstyles trained on large-scale dataset. This approach has proven to be both efficient and highly accurate in preserving strand details, which can be deployed as a generic hair prior to solve task-agnostic problems.

## 3 MODEL FORMULATION

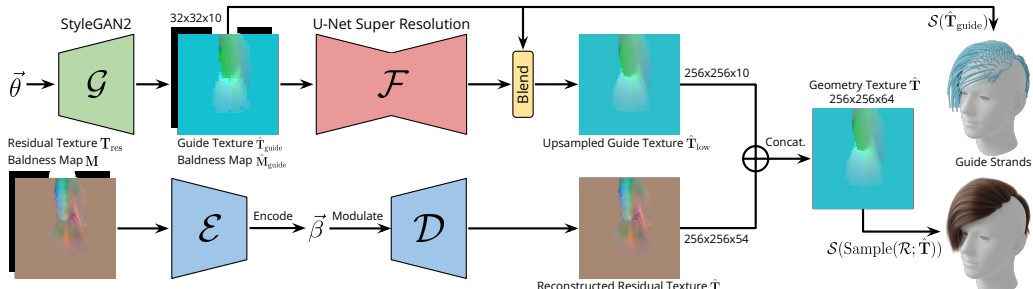

Figure 4: Architecture overview.

In PERM, hair is represented as geometry textures $\mathbf{T}$ storing the strand PCA coefficients $\vec{\gamma}$ (Sec. 3.1). These textures are decomposed into a lower-order component, the guide texture, which captures the global hair structure, and a higher-order component, the residual texture, which encodes local curl pattern information. To parameterize these textures, a parameter $\vec{\theta}$ is designed to generate the guide textures $\mathbf{T}_{\text{guide}}$ and $\mathbf{M}_{\text{guide}}$ through the function $\mathcal{G}(\vec{\theta})$, covering the sparse set of guide strands and the hair growing area on the scalp respectively (Sec. 3.2.1). These guide textures are upsampled with a deterministic function $\mathcal{F}(\cdot)$, whose output contains the complete hair structure but lacks high-frequency details (Sec. 3.2.2). To complement that, another parameter $\vec{\beta}$ is introduced to infer the residual texture $\mathbf{T}_{\text{res}}$ through the function $\mathcal{D}(\vec{\beta})$, storing high-frequency information such as different curl patterns (Sec. 3.2.3). Concatenating the upsampled guide texture and residual texture yields the geometry texture, which can then be sampled and decoded with the strand function $\mathcal{S}(\vec{\gamma})$ formulated in Sec. 3.1 to produce the final strand geometry. An overview of this architecture is illustrated in Fig. 4, and the resulting PERM model, denoted as $\mathcal{M}(\vec{\theta}, \vec{\beta}, \mathcal{R})$, can then be expressed as:

$$\mathcal{M}(\vec{\theta}, \vec{\beta}, \mathcal{R}; \Phi) = \mathcal{S}\Big(\text{Sample}\big(\mathcal{R}; \mathcal{F}(\mathcal{G}(\vec{\theta})) \oplus \mathcal{D}(\vec{\beta})\big)\Big), \tag{1}$$

where $\text{Sample}(\cdot)$ denotes nearest neighbor interpolation used to sample the synthesized texture with 2D root coordinates in the pre-defined root set $\mathcal{R}$, $\oplus$ represents the concatenation operation along the feature channels, and $\Phi$ specifies the full set of trainable parameters in PERM used to approximate the functions above. Once trained, parameters in $\Phi$ are held fixed, and novel 3D hairstyles are synthesized by varying $\vec{\theta}$ and $\vec{\beta}$ respectively, while the hair density can be adjusted by manipulating the number of roots in $\mathcal{R}$. The dimension of parameters $\vec{\theta}$ and $\vec{\beta}$ are both set to 512 in our model ($|\vec{\theta}| = |\vec{\beta}| = 512$). In the following, we delve into the details of each term in our model.

## 3.1 STRAND REPRESENTATION

For human hair, its cyclical growth behavior has been extensively documented in the biology community, where PCA has also been employed to analyze different hair types (Hoover et al., 2023; De La Mettrie et al., 2007; Loussouarn et al., 2007). Inspired by these biological findings, we define a low-dimensional parameter space based on PCA in the frequency domain. Formally, for 3D hair strands with a fixed number of points $\mathbf{S} = \{\mathbf{p}_1, \mathbf{p}_2, \ldots, \mathbf{p}_L\} \in \mathbb{R}^{L \times 3}$ ($L = 100$ in our experiments), they are represented as the output of a linear function $\mathcal{S}(\vec{\gamma})$ in the frequency domain:

$$\mathcal{S}(\vec{\gamma}; \mathbf{X}) = \mathrm{iDFT}(\bar{\mathbf{S}} + \vec{\gamma}^\top \mathbf{X}) = \mathrm{iDFT}(\bar{\mathbf{S}} + \sum_{n=1}^{|\vec{\gamma}|} \vec{\gamma}_n \mathbf{X}_n), \tag{2}$$

where $\vec{\gamma} = [\gamma_1, \ldots, \gamma_{|\vec{\gamma}|}]^\top$ is a vector of strand coefficients, and $\mathbf{X} = [\mathbf{X}_1, \mathbf{X}_2, \ldots, \mathbf{X}_{|\vec{\gamma}|}]^\top \in \mathbb{R}^{|\vec{\gamma}| \times 6k}$ forms a matrix of orthonormal principal components that capture phase variations in different strands, with $k = \lfloor L/2 \rfloor + 1$ referring the number of frequency bands. The term $\bar{\mathbf{S}} = [\bar{\mathbf{S}}_{\mathrm{real}}, \bar{\mathbf{S}}_{\mathrm{imag}}] \in \mathbb{R}^{k \times 3 \times 2}$ represents the mean phase vector of strands, and $\mathrm{iDFT}(\cdot)$ denotes the inverse Discrete Fourier Transform that maps strands back from the frequency domain to the spatial domain. In other sections, we often omit $\mathbf{X}$ in Eq. 2 for notational convenience.

To obtain quantities in Eq. 2, we first apply the Discrete Fourier Transform (DFT) along the $x$, $y$, $z$ axes to all strands in our collected hair models, and then compute the mean phase vector $\bar{\mathbf{S}}$ and solve the principal components $\mathbf{X}$ by performing PCA on the computed Fourier bases, which is similar to the prevalent PCA-based modeling methods in digital humans (Blanz & Vetter, 1999; Loper et al., 2015; Li et al., 2017). We set the number of PCA coefficients $|\vec{\gamma}| = 64$, which explained almost 100% of the variance in the training set (see Fig. 12).

Our PCA-based representation also facilitates a meaningful decomposition of strand geometry by forming an *interpretable* parameter space. Our analysis in Appendix B.4 reveals that the first 10 PCA coefficients are enough to effectively capture the global structure of each strand, and the remaining 54 coefficients encode high-frequency details, such as curl patterns (see Fig. 19). Leveraging this intuitive decomposition, we can effortlessly edit a given hairstyle or transfer the curl patterns from one hairstyle to another. Examples are provided in Fig. 6.

## 3.2 FULL HAIR REPRESENTATION

We define a 2D parameterization of hairstyles on the scalp surface as a regular $uv$ texture map, where each texel stores the strand PCA coefficients $\vec{\gamma}$ formulated in Eq. 2. These textures are referred to as hair geometry textures, denoted as $\mathbf{T} \in \mathbb{R}^{256 \times 256 \times 64}$. Similar to GroomGen (Zhou et al., 2023), we separately model the baldness area as a baldness map $\mathbf{M} \in \mathbb{R}^{256 \times 256}$. Both $\mathbf{T}$ and $\mathbf{M}$ are downsampled to $32 \times 32$ to represent guide strands (denoted as the guide texture $\mathbf{T}_{\mathrm{guide}}$ and mask $\mathbf{M}_{\mathrm{guide}}$), where only the low-rank PCA coefficients ($|\vec{\gamma}_{\mathrm{guide}}| = 10$) are kept in $\mathbf{T}_{\mathrm{guide}}$. As not all texels will be decoded to meaningful strands, $\mathbf{T}_{\mathrm{guide}}$ typically accommodates around 400 strands, as visualized in Fig. 1. Note that the concept of guide strands here is different from previous works (Zhou et al., 2023; Shen et al., 2023), where the guide strands are just quantitatively downsampled strands as visualized in Fig. 2 without the filtered structure.

Since guide strands only serve as a sparse representation of the full hair, an upsampling process is necessary to obtain the final hair strands. However, as illustrated in GroomGen, this process entails a *one-to-many* mapping (Zhou et al., 2023), wherein the same guide strands can be upsampled to yield diverse hairstyles with varying strand randomness and curliness. Achieving this property proves challenging, necessitating an orthogonal decomposition of the global hair structure and local curl patterns. Previous methods thus either downgraded to a deterministic one-to-one mapping (Sklyarova et al., 2023b) or resorted to a manual post-processing step (Zhou et al., 2023). To address this challenge, we further decompose our hair geometry texture $\mathbf{T}$ into $\mathbf{T}_{\mathrm{low}} \in \mathbb{R}^{256 \times 256 \times 10}$ and $\mathbf{T}_{\mathrm{res}} \in \mathbb{R}^{256 \times 256 \times 54}$ by splitting along the feature channels. Given the explainability of our strand PCA coefficients, we discern that $\mathbf{T}_{\mathrm{low}}$ encapsulates the global hair structure, while $\mathbf{T}_{\mathrm{res}}$ embodies the local curl patterns, denoted as the residual texture.

Leveraging this decomposition, we employ different neural networks to parameterize them, simulating the automated process of hair modeling pipeline.

### 3.2.1 GUIDE TEXTURE SYNTHESIS

Formally, for guide textures, we train a neural network to approximate the function, $\mathcal{G}(\vec{\theta}; \phi_1)$: $\mathbb{R}^{|\vec{\theta}|} \mapsto \mathbb{R}^{32 \times 32 \times (10+1)}$, which synthesizes $\mathbf{T}_{\text{guide}}$ and $\mathbf{M}_{\text{guide}}$ from the guide strand parameter $\vec{\theta}$, where $\phi_1$ denotes the trainable network parameters. In our formulation, we choose StyleGAN2 (Karras et al., 2020) as the backbone of our model, which has been proved to be a powerful generator in both 2D images and 3D feature tri-planes (Chan et al., 2022). It also ensures our guide strand parameter will follow the normal distribution $\vec{\theta} \sim \mathcal{N}(\mathbf{0}, \mathbf{I})$, making it easy to sample. Moreover, the intermediate $\mathcal{W}$ and $\mathcal{W}+$ spaces introduced in StyleGAN allow for more faithful latent embedding (Abdal et al., 2019), which in our case facilitates hair reconstruction either from 3D strands or 2D orientation images. Since our generator synthesizes both the guide texture $\mathbf{T}_{\text{guide}}$ and mask $\mathbf{M}_{\text{guide}}$, we use a similar dual discrimination method as EG3D (Chan et al., 2022), where we concatenate these two images and feed them into the discriminator. This operation ensures consistency between the generated texture and mask, which helps the generator to place zero-length strands in bald areas. Our training objective is identical to StyleGAN2, which consists of a non-saturating GAN loss (Goodfellow et al., 2014) with $R_1$ regularization (Mescheder et al., 2018) on both the texture and mask, where the regularization strengths are set to 5 and 1, respectively.

### 3.2.2 GUIDE TEXTURE UPSAMPLING

To upsample the synthesized guide textures, we formulate it as an image super resolution problem, which can be defined as the function, $\mathcal{F}(\mathbf{T}_{\text{guide}}, \mathbf{M}_{\text{guide}}; \phi_2) : \mathbb{R}^{32 \times 32 \times 11} \mapsto \mathbb{R}^{256 \times 256 \times 10}$, with trainable network parameters $\phi_2$. Note that this function operates as a deterministic mapping without parameter control, which aligns with both previous deep learning-based methods (Zhou et al., 2023; Sklyarova et al., 2023b) and the adaptive interpolation algorithm used in current hair modeling software. To approximate $\mathcal{F}(\cdot)$, we train a U-Net (Ronneberger et al., 2015) on the bilinearly upsampled textures, which translates them to 14-channel weight maps, where the first 4 channels represent the weight vector $\vec{\omega}$ for the 4 neighboring guide strands, and the last 10 channels represent a residual vector $\vec{\delta}$ to correct the blended coefficients. Therefore, the final coefficient $\vec{\gamma}_{\text{low}}$ can be calculated as:

$$\vec{\gamma}_{\text{low}} = \vec{\omega}^\top \boldsymbol{\gamma}_{\text{guide}} + \vec{\delta} = \sum_{n=1}^{4} \omega_n \vec{\gamma}_{\text{guide}}^n + \vec{\delta}. \qquad (3)$$

Our experiments demonstrate that this weight-based blending output allows the network to converge to a sharper result compared to predicting coefficients directly.

We train the U-Net in a supervised manner, where we employ both an $L_1$ loss for the blended texture $\hat{\mathbf{T}}_{\text{low}}$ and a geometric loss $\mathcal{L}_{\text{geo}}$ for the decoded strand geometry. Specifically, the geometric loss $\mathcal{L}_{\text{geo}}$ encompasses an $L_1$ loss for the point position $\hat{\mathbf{p}}_n$, a cosine distance for the orientation $\hat{\mathbf{d}}_n = \hat{\mathbf{p}}_{n+1} - \hat{\mathbf{p}}_n$ (Rosu et al., 2022), and an $L_1$ loss for the curvature, represented as the $L_2$ norm of binormal vector $\hat{\mathbf{b}}_n = \|\hat{\mathbf{d}}_n \times \hat{\mathbf{d}}_{n+1}\|_2$ (Sklyarova et al., 2023a):

$$\mathcal{L}_{\text{geo}} = \sum_{n=1}^{L} \|\hat{\mathbf{p}}_n - \mathbf{p}_n\|_1 + (1 - \hat{\mathbf{d}}_n \cdot \mathbf{d}_n) + \|\hat{\mathbf{b}}_n - \mathbf{b}_n\|_1. \qquad (4)$$

A regularization term is included as well to constrain the residual vector $\vec{\delta}$ towards 0. The overall loss function, denoted as $\mathcal{L}_{\text{superres}}$, then can be expressed as:

$$\mathcal{L}_{\text{superres}} = \lambda_{\text{tex}} \|\hat{\mathbf{T}}_{\text{low}} - \mathbf{T}_{\text{low}}\|_1 + \lambda_{\text{geo}} \mathcal{L}_{\text{geo}} + \lambda_{\text{reg}} \|\vec{\delta}\|_2^2, \qquad (5)$$

where the weighting factors $\lambda_{\text{tex}}$, $\lambda_{\text{geo}}$ and $\lambda_{\text{reg}}$ are set to 1, 1 and 0.1, respectively.

### 3.2.3 RESIDUAL TEXTURE SYNTHESIS

To simulate the artistic process of creating different curl patterns, we train a neural network to approximate the function, $\mathcal{D}(\vec{\beta}; \phi_3) : \mathbb{R}^{|\vec{\beta}|} \mapsto \mathbb{R}^{256 \times 256 \times 54}$, where we learn to synthesize the residual texture $\mathbf{T}_{\text{res}}$ from the hair styling parameter $\vec{\beta}$, and $\phi_3$ represents the trainable network parameters.

Although StyleGAN2 performs well in generating textures for guide strands, we found it struggles with residual textures, as they contain more data than high-resolution RGB images. Therefore, we

opted for VAE (Kingma & Welling, 2013) as the backbone, where the encoder adopts an architecture similar to pSp (Richardson et al., 2021), and the decoder takes the same architecture as StyleGAN2 generator. Essentially, the encoder $\mathcal{E}$ projects the residual texture $\mathbf{T}_{\text{res}}$ and baldness map $\mathbf{M}$ into the latent space, where the latent vectors will then be used to modulate the decoder to reconstruct the input. A similar loss function as Eq. 5 is defined as the training objective for the VAE, which can be expressed as:

$$\mathcal{L}_{\text{res}} = \lambda_{\text{tex}}(\|\hat{\mathbf{T}}_{\text{res}} - \mathbf{T}_{\text{res}}\|_1 + \|\hat{\mathbf{M}} - \mathbf{M}\|_1) + \lambda_{\text{geo}}\mathcal{L}_{\text{geo}} + \lambda_{\text{KL}}\mathcal{L}_{\text{KL}}, \quad (6)$$

where the reconstruction terms are computed on the residual texture, baldness map, and decoded strand geometry. The weighting factors $\lambda_{\text{tex}}$, $\lambda_{\text{geo}}$, and $\lambda_{\text{KL}}$ are set to 10, 1, and $1e^{-4}$, respectively.

## 4 EXPERIMENTS

We train PERM on an augmented version of USC-HairSalon (Hu et al., 2015), which contains a total of $21,054$ data samples. For evaluation, we compiled a separate dataset of 17 publicly available hair models, comprising $151,829$ strands in total. Detailed data augmentation and source of our testing data are provided in Appendix B.1. Since USC-HairSalon lacks sufficient curly hair data, we curated a private dataset consisting of 80 manually groomed hairstyles, comprising a total of $4,368,679$ strands with a greater diversity of style. We conducted experiments on this private dataset to further validate the robustness of our representation, whose results are detailed in Appendix B.4.

### 4.1 STRAND REPRESENTATION

We first conducted extensive experiments to evaluate our PCA-based strand representation against various alternative deep learning-based representations and a simpler PCA-based formulation without DFT. Contrary to the common belief that neural network methods outperform linear models like PCA, our experiments show that the PCA method performs better in the context of hair modeling.

To quantitatively assess their reconstruction capabilities, we present the mean position error (pos. err.), computed as the average Euclidean distance between corresponding points on the reconstructed strands and the ground truth. Additionally, we report the mean curvature error (cur. err.), defined as the $L_1$ norm between the curvatures of reconstructed and ground truth strands. Please refer to Appendix B.3 for details of the metrics.

Table 1: Reconstruction errors reported on different strand representations. Here **boldface** corresponds to the best result and underline means the second best.

|  | Freq. VAE | CNN VAE | Transformer VAE | MLP VAE | PCA | Freq. PCA (Ours) |
|---|---|---|---|---|---|---|
| # params. | 15.67M | 759.24K | 2.29M | 15.67M | 19.31K | 19.50K |
| pos. err. | 1.211 | 0.446 | 0.288 | 0.158 | **0.019** | 0.020 |
| cur. err. | **0.910** | 6.250 | 8.539 | 1.594 | 2.361 | 2.150 |

Quantitative results are reported in Table 1, where the deep learning-based strand representations include a VAE model trained with frequency features (Freq. VAE) (Zhou et al., 2023), and other VAE variants with CNN, Transformer (Vaswani et al., 2017), and ModSIREN MLP (Mehta et al., 2021) decoders. All configurations compress strands to 64-dimensional vectors. While simple, our PCA-based representations demonstrate a remarkably small position error, achieving this with a significantly reduced number of parameters compared to all different VAE variants. It also significantly reduces training time and GPU memory consumption due to its analytical computation. Note that position error is more dominant in determining the quality of reconstruction. Although some VAEs may achieve a lower curvature error, their reconstructed strands often appear noticeably different from the ground truth because they fail to preserve the overall shape (see Fig. 5). To further demonstrate the robustness of our strand representation, we trained all models on the curlier private dataset and evaluated them using the same 17 publicly available hairstyles. Detailed experimental results can be found in Appendix B.4, which reveal a trend similar to those in Table 1 (our PCA-based representation achieves the lowest position error and a comparatively low curvature error).

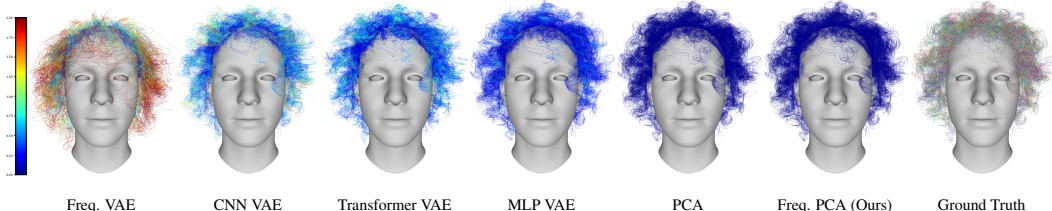

| Freq. VAE | CNN VAE | Transformer VAE | MLP VAE | PCA | Freq. PCA (Ours) | Ground Truth |

Figure 5: Comparison of our PCA-based strand representation (Freq. PCA) with other VAE and PCA-based representations. Reconstructed strands are color-coded by their position error.

Additionally, with our proposed PCA-based strand representation, we can smooth a given hairstyle by truncating its strand PCA coefficients (Fig. 6a, b), or transfer hairstyle details by blending the low-rank and high-rank portions of PCA coefficients from different hairstyles (Fig. 6c – f).

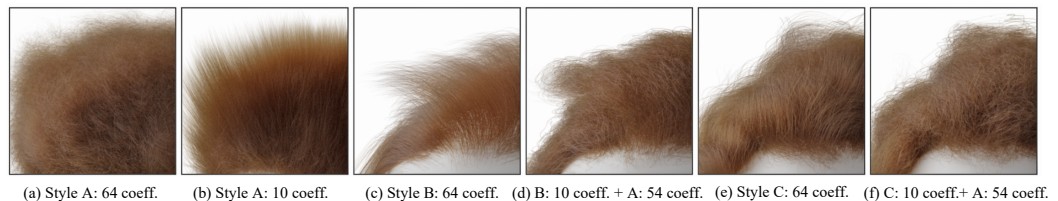

| (a) Style A: 64 coeff. | (b) Style A: 10 coeff. | (c) Style B: 64 coeff. | (d) B: 10 coeff. + A: 54 coeff. | (e) Style C: 64 coeff. | (f) C: 10 coeff.+ A: 54 coeff. |

Figure 6: Examples of hairstyle editing with our PCA-based strand representation. (a) and (b) demonstrate hair smoothing by truncating the strand PCA coefficient from 64 to 10. (c) – (f) show the detail transfer between different hairstyles.

## 4.2 FULL HAIR REPRESENTATION

**3D Hair Parameterization**  To evaluate our proposed hairstyle representation, we first fit PERM parameters to target 3D hair models through differentiable optimization, which we term as *3D hair parameterization*. Technical details are provided in Appendix B.5. In Fig. 8 we present a subset of fitted results that illustrate our model's capability to accurately recover the given 3D hair models. Even if the target hair has no guide strands, our model can generate reasonable guide strands to depict their overall shapes, obtaining directly from $\mathcal{S}\big(\mathcal{G}(\vec{\theta}^{*})\big)$.

**Comparison with Groom-Gen**  The most relevant previous work is Groom-Gen (Zhou et al., 2023) that learns hierarchical representations of 3D hair. As there is no publicly available GroomGen code, we implemented and trained it on the same augmented USC-HairSalon dataset. We also contacted the authors to obtain part of their official checkpoints to

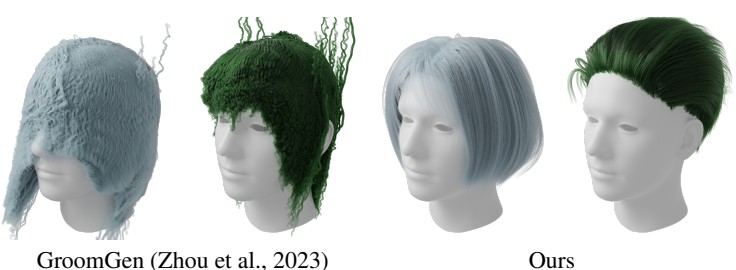

GroomGen (Zhou et al., 2023)          Ours

Figure 7: Comparison with our implementation of GroomGen (Zhou et al., 2023) on random hairstyle synthesis. Full results are available in Fig. 27.

verify the correctness of our implementation. Detailed verification can be found in Appendix C.4, where we compared each network module with the available official checkpoints to demonstrate the correctness of our implementation. Through experiments with our implementation, we found the neural upsampler module in GroomGen, which is a GAN architecture, is very unstable to train, frequently leading to collapsed results. In Fig. 7 we visualize randomly synthesized hairstyles by sampling our parameter space and GroomGen's latent space with the same Gaussian noise, demonstrating their collapsed output with weird shapes and curls. We further conducted a quan-

titative evaluation of reconstruction errors on the 17 testing hairstyles, where our model achieves a position error of 1.658 and curvature error of 0.769, both of which are lower than GroomGen's corresponding values (position error 2.570, curvature error 6.199). During quantitative evaluation, we also found that 100 points per strand are insufficient to faithfully represent long kinky hairstyles. Examples are provided in Fig. 25.

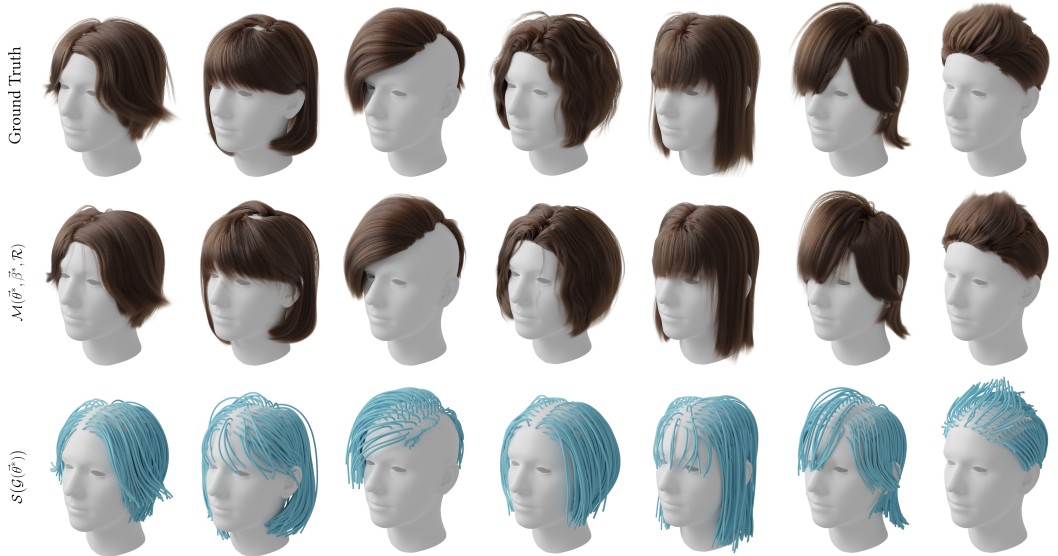

Figure 8: A subset of 3D hair models fitted by PERM.

## 5 APPLICATIONS

**Single-view Hair Reconstruction** With 2D observations as input, we can reconstruct 3D hair by optimizing PERM parameters to minimize the energy of generated hair when projected to the 2D images. To fit 3D hair to 2D supervisions, we design a differentiable rendering pipeline, as illustrated in Fig. 9. In the pipeline, we pre-compute the transformation between our in-house head mesh and SMPL-X (Pavlakos et al., 2019), denoted as the function $\mathcal{T}$, thus placing strand polylines generated from PERM onto SMPL-X. We then attach a cylinder mesh onto each strand segment, as well as computing the 3D orientation of each vertex as the per-vertex feature. These features are projected and rendered using Nvdiffrast (Laine et al., 2020) with the estimated camera parameters, thereby obtaining the rendered hair mask $\mathbf{M}_{render}$ and strand map $\mathbf{O}_{render}$. By computing the pixel-wise mask loss and strand

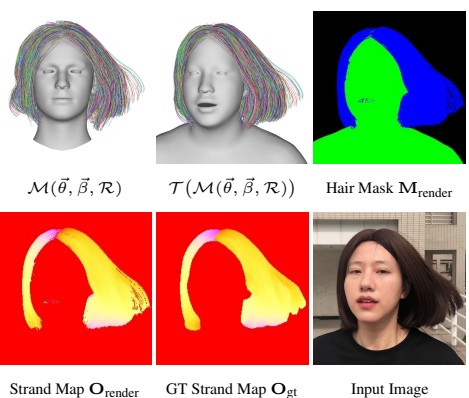

Figure 9: Illustration of our differentiable rendering pipeline for single-view reconstruction.

map loss similar to HairStep (Zheng et al., 2023), gradients are back-propagated to optimize $\vec{\theta}$ and $\vec{\beta}$ with decoupled weight decay regularization (Loshchilov & Hutter, 2017). A penetration loss is applied as well to penalize hair intersecting the body geometry.

In Fig. 10 and Fig. 30, we qualitatively compare our single-view reconstruction results with HairStep (Zheng et al., 2023), the state-of-the-art open-source method for single-view hair reconstruction. With PERM as a robust prior to ensure the generation quality, our method not only excels at reconstructing the global structure of the reference hairstyle but also better preserves local curl patterns compared to HairStep. With the introduction of penetration loss, our algorithm effectively handles input images with large and tilted head poses, avoiding outputs with unintended bald areas,

an artifact we occasionally observed in HairStep. In Fig. 31 we show the single-view reconstruction results of HAAR (Sklyarova et al., 2023b), a recent text-conditioned hair generation method. However, since textual descriptions cannot capture the intricate details of a hairstyle, its output can only produce a rough approximation.

**PERM Parameter Editing**   As we introduce separate parameters to control the global hair shape and local strand details, we can edit the hairstyle with varying granularity by either swapping the parameters from different hairstyles or interpolating the corresponding parameters. In the last row of Fig. 30, we edit our reconstructed hairstyles by swapping their $\vec{\beta}$ parameters from a wavy reference hairstyle, thereby altering their curl patterns while maintaining a similar haircut. More examples of hairstyle interpolation are provided in Appendix D.2, with comparisons to (Weng et al., 2013) and (Zhou et al., 2018).

**Hair-conditioned Image Generation**   Latest text-to-image (T2I) models (e.g., Adobe Firefly (Adobe, 2024)) can generate high-quality portrait images. However, the text embedding of simple prompts like *"wavy and short hair"* is not precise enough to represent a specific hairstyle. To tackle this issue, we show the use of PERM in facilitating conditional image generation. Specifically, We could use PERM to sample and edit hairstyles in 3D, or reconstruct 3D hairstyles from images, and then feed the depth and edge information extracted from the hair geometry to the T2I models to generate the final images. As shown in Fig. 32, pure text prompts often lead to images with different hairstyles, while combined with the input hair reference, the generated images show a much more consistent hairstyle. More examples are available in Fig. 33.

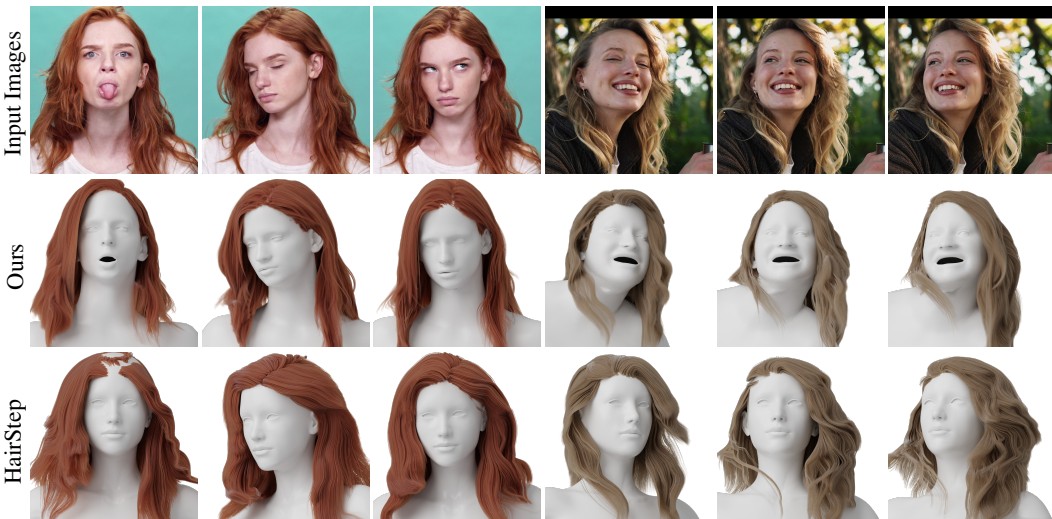

Figure 10: Single-view hair reconstruction on images with curly hair and tilted head poses, with comparison to HairStep (Zheng et al., 2023). More examples are available in Fig. 30.

## 6   CONCLUSION

We present PERM, a parametric hair representation supporting independent control over the global hair structure and local curl patterns. While our approach demonstrates strong performance when trained on 3D hair models, exploring methods to develop a 3D generative model for hair from 2D in-the-wild images remains a compelling direction for future research. Besides, since our $uv$-based hair representation is compatible with the existing diffusion pipeline, with more data available, it would be interesting to train a 3D diffusion model for full-head synthesis with controllable hair modeling. Last but not least, aligning with trends in other 2D or 3D generative tasks, controllable 3D hair synthesis with multi-modal input signals presents another exciting avenue for future work in this domain.

## ACKNOWLEDGMENTS

The authors would like to thank Yujian Zheng for providing the source code and model of HairStep, and for discussing the HairStep results in the paper with us. We also extend our sincere gratitude to Yao Feng for assisting us in setting up DELTA for face and shoulder reconstruction, and Hao Li for contributing a selection of the realistic head and hair assets used in this research.

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

# A FORMULATION DETAILS

## A.1 HAIR GEOMETRY TEXTURES

Unlike triangle meshes that share the same topology between different bodies and faces, hair tends to have a different number of strands for different hairstyles, making traditional PCA-based blend shapes impossible on these data. Although we can force different hairstyles to have the same number of strands by introducing a resampling step, the computed blend shapes would still have a fixed number of strands, which are less flexible in real applications. For flexibility, we store each hair model as a 2D texture map of strand PCA coefficients. With the scalp parameterization proposed by Wang et al. (2009), we are able to unwarp the 3D scalp surface to a 2D $uv$ plane. However, naively storing strand PCA coefficients on the projected 2D root positions will cause two problems: (1) different strands may be projected to the same texel due to discreterization, thus causing collision problems; (2) some texels may receive no strands, thus leaving missing values in the projected textures.

To address these issues, we fit our geometry textures with two steps: first, we find the nearest 2D hair root for each texel, and store the corresponding strand PCA coefficients at that texel. For those texels whose distance to its nearest hair root is above a threshold $\epsilon = 0.01$, we store a special vector that will be decoded to strands with zero length, and mark those texels as a baldness map $\mathbf{M}$ (Zhou et al., 2023). This step ensures no missing values in the texture, which we denote as the initialized geometry texture $\mathbf{T}_{\text{init}}$. Both $\mathbf{M}$ and $\mathbf{T}_{\text{init}}$ have the resolution $256 \times 256$, which is empirically set considering the trade-off between size and expressiveness. To ensure that geometry textures can properly recover the original 3D hairstyle, we optimize them directly in the second step:

$$\mathbf{T}^* := \arg\min_{\mathbf{T}} \mathcal{L}_{\text{geo}}\Big(\mathcal{S}\big(\text{Sample}(\mathcal{R}; \mathbf{T})\big)\Big), \qquad (7)$$

where we use loss $\mathcal{L}_{\text{geo}}$ to measure the reconstruction difference on strand geometry.

We employ the Adam optimizer (Kingma & Ba, 2014) with a learning rate of 0.001. With $\mathbf{T}_{\text{init}}$ as initialization, our experiments show that the optimization process converges within 500 iterations, and in Fig. 11 we show an example of the original hairstyle, the fitted geometry texture, and the recovered 3D hairstyles by sampling and decoding from the texture with different numbers of roots. These fitted geometry textures finally form a unified representation across different hairstyles, which have the same size $256 \times 256 \times 64$ and allow for arbitrary sampling.

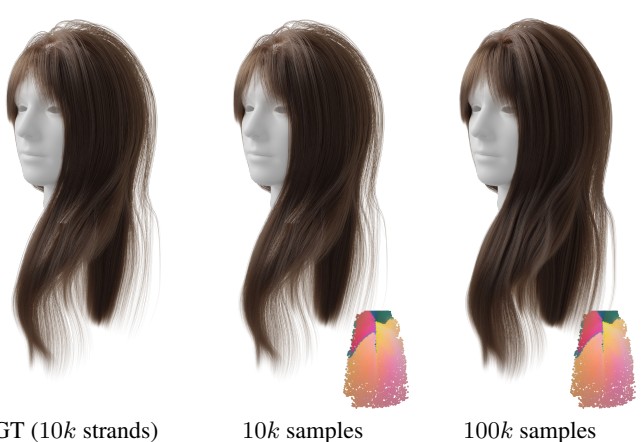

GT (10$k$ strands)   10$k$ samples   100$k$ samples

Figure 11: Illustration of fitted hair geometry textures and the corresponding 3D hairstyle. Here baldness maps are visualized as the alpha channel in the RGBA textures.

## A.2 NETWORK ARCHITECTURE

**StyleGAN2 Backbone** Our StyleGAN2 backbone follows the official implementation of (Karras et al., 2020)[1], with a mapping network of 4 hidden layers. We modify the output convolutions such that they produce a feature image of shape $32 \times 32 \times 10$. Subsequently, a small MLP decoder is employed to map the output features to 10-dimensional strand PCA coefficients and a single scalar for guide mask. The MLP decoder consists of a single hidden layer of 64 hidden units and uses the softplus activation function. Note that we do not utilize pre-trained StyleGAN2 checkpoints for our task, the entire module is trained from scratch.

---

[1] https://github.com/NVlabs/stylegan2-ada-pytorch

**U-Net Super Resolution**    Our U-Net module is implemented based on an unofficial online implementation[2], which translates the bilinearly upsampled textures of shape $256 \times 256 \times 11$ to weight maps of shape $256 \times 256 \times 14$. The convolution layers progressively downsample the input to a shape of $16 \times 16 \times 512$, which is followed by $4$ bilinear upsampling and double convolution layers with skip connections to produce the weight map.

**VAE**    In our VAE module, the encoder adopts a similar architecture to pSp (Richardson et al., 2021), which contains $4$ IR-SE blocks (Hu et al., 2018) to extract a feature image of shape $4 \times 4 \times 512$. This feature image is then flattened and processed through a fully-connected layer to derive the mean and variance of $\vec{\beta}$ of $512$ dimensions. The decoder mirrors the StyleGAN2 generator but omits its mapping network. Its output size is modified to $256 \times 256 \times 54$, followed by the same MLP decoder that maps the output features to $54$-dimensional strand PCA coefficients and a single scalar for baldness map.

## B    Experiment Details

### B.1    Datasets

We train PERM on USC-HairSalon (Hu et al., 2015), which is a dataset comprising 343 3D hair models collected from online game communities. To increase diversity, we employ the data augmentation method proposed in HairNet (Zhou et al., 2018), where different hair models within the same style class are blended to produce novel hairstyles. The blended hairstyles are further augmented by horizontal flipping, resulting in a total of $21,054$ data samples used for training.

To assess the performance of our model, we compiled a dataset of 3D hair models from various publicly available resources, including CT2Hair (Shen et al., 2023) (10 hairstyles), StructureAware-Hair (Luo et al., 2013) (3 hairstyles), and Cem Yuksel's website[3] (4 hairstyles). We preprocess these data to have the same number of points ($L = 100$) on each strand, and register them onto the same head mesh.

### B.2    Training Details

We train our model and conduct all experiments on a desktop machine with an Intel® Core™ i9-10850K CPU @ 3.60GHz, 64GB memory, and an NVIDIA RTX 3090 GPU. Our code is implemented with Python 3.9.18, PyTorch 1.11.0, and CUDA Toolkit 11.3.

In our model, each network module is trained separately using the Adam optimizer (Kingma & Ba, 2014). The StyleGAN2 backbone has a learning rate of $0.002$ for its generator and $0.001$ for its discriminator, leading to a stable training configuration in our case. The StyleGAN2 backbone is trained for $3,000$K images with a batch size of $4$, which takes around $1$ day on our machine. For both the U-Net and VAE, we set their learning rates to $0.002$ and train them for $2,000$K images with a batch size of $4$, each taking around $1$ day on our machine.

### B.3    Quantitative Metrics

To quantitatively measure the reconstruction capability of our model, we first report the mean *position error* (pos. err.), which is essentially the average Euclidean distance between the corresponding points on the reconstructed strands and the ground truth. We further report the mean *curvature error* (cur. err.) that measures the $L_1$ norm between the curvatures of reconstructed and ground truth strands, where the curvature is defined as the reciprocal of the circumradius of $3$ consecutive points $\mathbf{p}_{i-1}$, $\mathbf{p}_i$, and $\mathbf{p}_{i+1}$ on the strand, which can be computed as:

$$\mathrm{cur}(\mathbf{p}_i) = \frac{2\|(\mathbf{p}_{i-1} - \mathbf{p}_{i+1}) \times (\mathbf{p}_i - \mathbf{p}_{i+1})\|}{\|\mathbf{p}_{i-1} - \mathbf{p}_{i+1}\| \cdot \|\mathbf{p}_i - \mathbf{p}_{i+1}\| \cdot \|\mathbf{p}_{i-1} - \mathbf{p}_i\|}. \tag{8}$$

---

[2]https://github.com/milesial/Pytorch-UNet
[3]http://www.cemyuksel.com/research/hairmodels/

### B.4 PCA-BASED STRAND REPRESENTATION

In Fig. 12, we illustrate the explained cumulative relative variance against the number of principal components. Although 20 PCA coefficients appear to capture nearly 100% of the variance in the *training set*, increasing the number of coefficients improves the generalizability of our representation to unseen data, as evidenced by the reconstruction errors shown in Fig. 13 and Fig. 14 on the *testing set* with 17 public hairstyles. We suspect that though high-frequency features are sparse and contribute little to the variance, they are important to generalize fitted PCs to unseen curlier data. Considering this issue, we opt for 64 principal components, a choice consistent with most previous work on strand representation (Rosu et al., 2022; Sklyarova et al., 2023a; Zhou et al., 2023), while achieving significantly lower reconstruction errors.

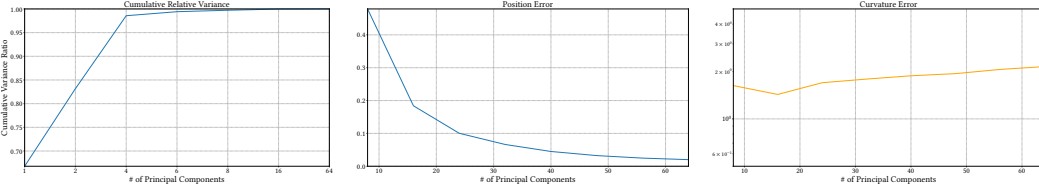

Figure 12: Cumulative relative variance.     Figure 13: Position error.     Figure 14: Curvature error.

To better elaborate the difference between PCA and Freq. PCA, we visualize the reconstructed hairstyles of these two representations in Fig. 15, where strands are color-coded by their curvature error. The average curvature errors for the entire hairstyle of these two representations are 2.671 and 2.341, respectively. These results demonstrate the clear advantage of our Freq. PCA representation in preserving strand curvature, both qualitatively and quantitatively.

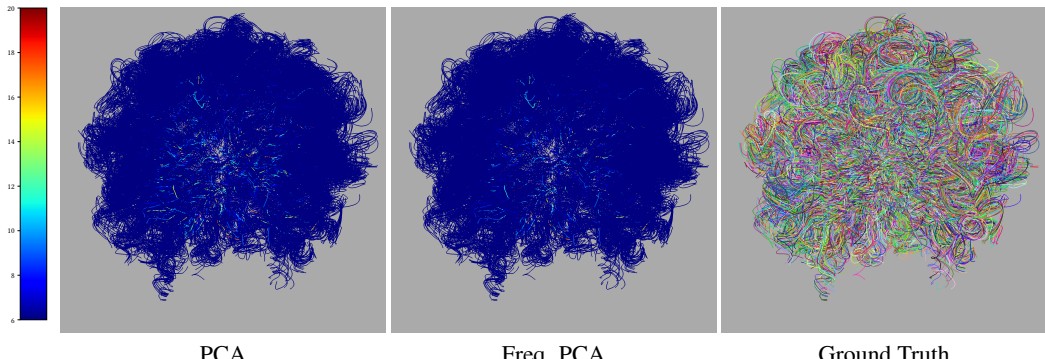

PCA                          Freq. PCA                          Ground Truth

Figure 15: Comparison of PCA and Freq. PCA-based strand representation. Reconstructed strands are color-coded by their curvature error.

To further demonstrate the robustness of our strand representation, we trained all models on a dataset of 80 manually groomed hairstyles, with curliness types varying from I to VI as defined in (Loussouarn et al., 2007), comprising a total of 4, 368, 679 strands. The newly trained models are then tested on the same 17 publicly available hairstyles, with reconstruction errors reported in Table 2. Even on this distinct dataset, our strand representation consistently achieves a significantly lower position error compared to other deep learning-based representations.

Similarly, we plot the cumulative relative variance, reconstructed position error and curvature error as a function of the number of principal components in Fig. 16, Fig. 17, and Fig. 18. The overall trends in these figures closely align with the results we observed in our previous experiments.

In Fig. 19 we compare the reconstruction of the same hairstyle using different numbers of strand PCA coefficients. 64 coefficients per strand are sufficient to closely replicate the original hair, and its curl patterns are captured down to 15 coefficients per strand. While smooth, 5 coefficients per strand fail to represent the global hair structure, whereas 10 coefficients per strand provide the most

Table 2: Reconstruction errors reported on strand representations trained on a different strand dataset. Here **boldface** corresponds to the best result and underline means the second best.

| | Freq. VAE | CNN VAE | Transformer VAE | MLP VAE | PCA | Freq. PCA (Ours) |
|---|---|---|---|---|---|---|
| # params. | 15.67M | 759.24K | 2.29M | 15.67M | 19.31K | 19.50K |
| pos. err. | 1.450 | 0.142 | 0.294 | 0.117 | **0.026** | **0.026** |
| cur. err. | **1.409** | 4.125 | 10.331 | 2.120 | 2.490 | 2.364 |

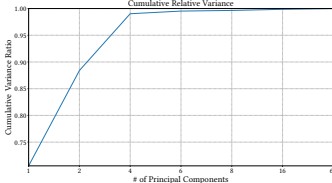
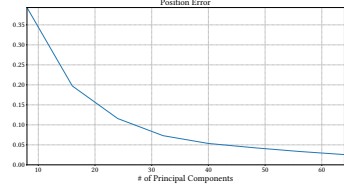
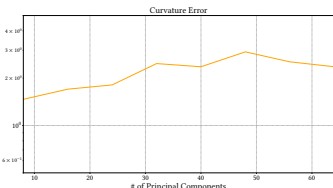

Figure 16: Cumulative relative variance.  Figure 17: Position error.  Figure 18: Curvature error.

balanced output in terms of smoothness and global structure preservation. Based on this observation, we choose to use 10 coefficients to capture the global structure of each strand, leaving high-frequency details to the remaining 54 coefficients.

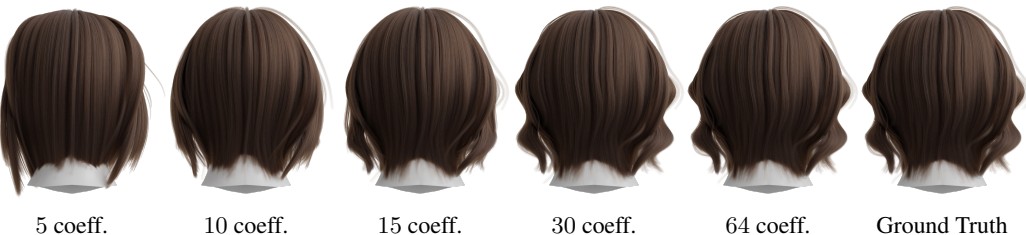

| 5 coeff. | 10 coeff. | 15 coeff. | 30 coeff. | 64 coeff. | Ground Truth |

Figure 19: Comparison of the same hairstyle reconstructed with different numbers of strand PCA coefficients.

## B.5 3D HAIR PARAMETERIZATION

To fit PERM parameters to target 3D hair models, we formulate it as an optimization problem, where the objective is defined as:

$$\vec{\theta}^*, \vec{\beta}^* := \arg\min_{\vec{\theta}, \vec{\beta}} \|\mathcal{F}(\mathcal{G}(\vec{\theta})) \oplus \mathcal{D}(\vec{\beta}) - \mathbf{T}\|_1 + \mathcal{L}_{\text{geo}}. \tag{9}$$

We employ the Adam optimizer (Kingma & Ba, 2014) with an initial learning rate of $0.1$ and a cosine annealing schedule for the learning rate. For better convergence, we first optimize $\vec{\theta}$ only for $1,000$ iterations as a warm-up to match the global shape, and then jointly optimize $\vec{\theta}$ and $\vec{\beta}$ for $4,000$ iterations.

With PERM, we fit parameters to hundreds of publicly available 3D hair models sourced from the Internet. Similar to AMASS (Mahmood et al., 2019), we curated a dataset of 3D hair in a unified and parametric manner, which we released as well to facilitate future research.

## B.6 BASELINES

**GroomGen (Zhou et al., 2023)**   As the authors of GroomGen have not publicly release their code, we implement it ourselves with Python 3.9.18 and Pytorch 1.11.0. The model is then trained on the same USC-HairSalon dataset as described in Appendix. B.1. Our implementation is further verified with part of the official checkpoints we obtained from the authors.

**HairStep (Zheng et al., 2023)**   We use the pre-trained HairStep model released by the authors[4].

**Strand VAEs**   The architecture of different strand VAEs compared in Table 1 is adapted from GroomGen (Zhou et al., 2023), where we only modify the decoder architectures to adopt CNN, Transformer (Vaswani et al., 2017), and ModSIREN (Mehta et al., 2021).

## C   ABLATION STUDY

### C.1   GUIDE TEXTURE SYNTHESIS

To assess the performance of StyleGAN2 in guide texture synthesis, we first create a PCA-based representation for guide textures with a similar formulation as our strand representation, and set the subspace dimension to 512 to align with PERM's setup. We then embed guide textures from the testing set into these latent spaces, followed by a decoding step to reconstruct them. Strands are further decoded from the reconstructed textures to evaluate quantitative errors, which are reported in Table 3. It is evident that results from StyleGAN2 exhibit both smaller position and curvature errors, indicating that StyleGAN2 learns a more expressive latent space than PCA. Fig. 20 displays examples of reconstructed guide strands, illustrating that StyleGAN2 more faithfully preserves the structure of the ground truth.

Table 3: Reconstruction errors reported on different architectures for guide textures.

|  | **Guide Textures** | |
|---|---|---|
|  | PCA | StyleGAN2 (Ours) |
| pos. err. | 1.163 | **0.611** |
| cur. err. | 2.939 | **1.582** |

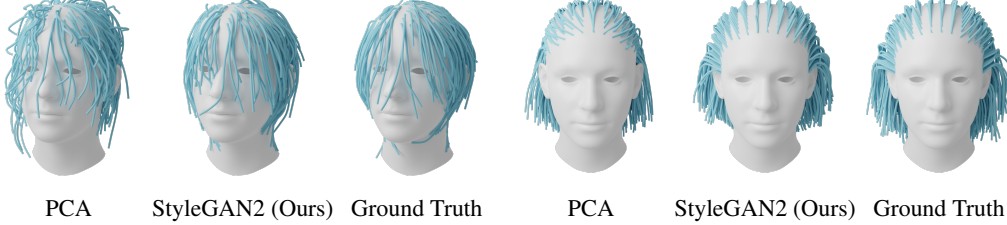

| PCA | StyleGAN2 (Ours) | Ground Truth | PCA | StyleGAN2 (Ours) | Ground Truth |

Figure 20: Comparison of PCA and StyleGAN2 in reconstructing given guide strands.

Reasons for the failure of PCA are straightforward: USC-HairSalon contains a vast number of strands, specifically $3,150,559$, for us to learn the PCA subspace, with a compression rate of approximately $78.7\%$. However, for guide textures, we can only create $21,054$ samples after data augmentation, which is merely around $0.6\%$ of the total number of available strands. Moreover, the compression rate increases to $95\%$. These two factors lead to the poor performance of PCA in this case.

### C.2   GUIDE TEXTURE UPSAMPLING

In Fig. 21 we compare the strands decoded from textures upsampled with different interpolation methods. Nearest neighbor interpolation (2nd column) produces aliased strands as it involves simple repetition. Bilinear interpolation (3rd column) leads to smoother strands, but introduces unwanted flyaway fibers, particularly in the forehead area. Our method (5th column) achieves the most natural result, which resembles the shape of guide strands and forms reasonable hair partitions. In addition, we employ the same U-Net architecture but configure it to directly predict strand PCA coefficients rather than the blending weights. The output (4th column) contains more flyaway strands and a reduced fidelity in capturing the overall hair shape, particularly noticeable in the lower part of the hair.

---

[4]https://github.com/GAP-LAB-CUHK-SZ/HairStep

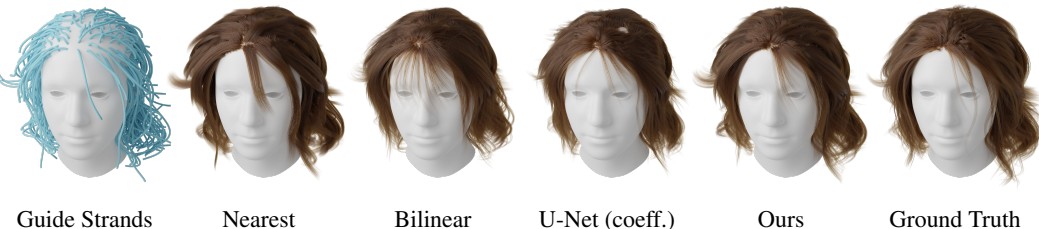

| Guide Strands | Nearest | Bilinear | U-Net (coeff.) | Ours | Ground Truth |

Figure 21: Comparison of strands upsampled with different interpolation methods.

## C.3 RESIDUAL TEXTURE SYNTHESIS

In Fig. 22 we evaluate various network architectures for residual texture synthesis. When residual textures are omitted, the resulting hairstyle captures only the global structure of the ground truth, lacking all high-frequency details such as different curl patterns (1st column). Though StyleGAN2 performed well in synthesizing guide strands, we found it collapsed to produce blurry residual textures when embedding them into the latent space, as shown in Fig. 22 (2nd column). We conjecture that this issue may be attributed to the large size of residual textures ($256 \times 256 \times 54$), as they contain more data than high-resolution RGB images ($1024 \times 1024 \times 3$). Moreover, our dataset is smaller than those image datasets used for StyleGAN training (e.g., FFHQ (Karras et al., 2019), which contains 70K high-quality portrait images), thus further suppressing the expressiveness of StyleGAN2. Considering these constraints, we devised our network architecture as a VAE, trading some sampling diversity for higher reconstruction fidelity, which reconstructs sharper residual textures (3rd column).

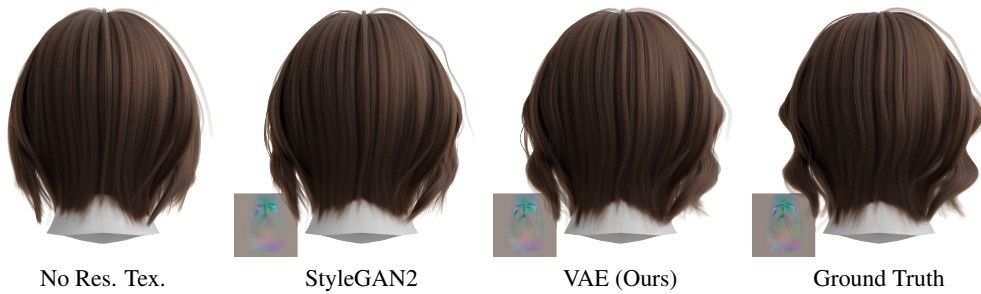

| No Res. Tex. | StyleGAN2 | VAE (Ours) | Ground Truth |

Figure 22: Residual texture reconstruction with different network architectures. Insets show the visualization of the corresponding residual texture. Please zoom in for details.

## C.4 ANALYSIS OF GROOMGEN

Since GroomGen (Zhou et al., 2023) is not fully open-sourced, we first verify the correctness of our implementation by comparing it to the official checkpoints of the strand VAE and hairstyle VAE, which we obtained from the authors.

For the strand VAE, we conducted the same strand reconstruction experiments as described in Sec. 4.1 and visualize the reconstructed results in Fig. 23, where both our implementation and the official version struggle to preserve the overall structure of the given hairstyle. From a quantitative perspective, our implementation even outperformed the official model, achieving lower position and curvature errors on the test data. Specifically,

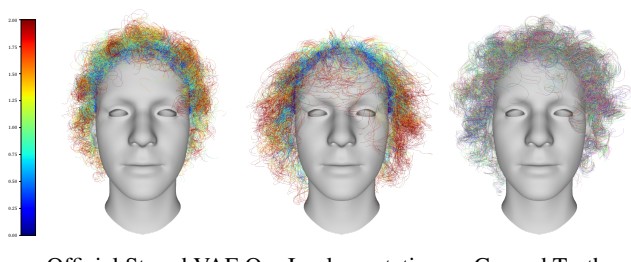

Official Strand VAE   Our Implementation   Ground Truth

Figure 23: Comparison of our strand VAE implementation with the official checkpoint of GroomGen (Zhou et al., 2023).

the official checkpoint yielded a position error of 1.521 and a curvature error of 1.337, whereas our implementation achieved 1.211 and 0.910, respectively.

For the hairstyle VAE, we were unable to embed guide strands from our head mesh into Groom-Gen's latent space due to the lack of $uv$ parameterization of the head mesh they used. Instead, we randomly sampled the latent spaces of both our implementation and the official version, using the same Gaussian noise, and visualized the decoded guide strands in Fig. 24. As expected, our implementation generates natural-looking guide strands, though they are sometimes overly smooth. In contrast, the official hairstyle VAE produces curlier guide strands, but sometimes, the results appear less natural. We hypothesize that these differences arise from the difference of the training data.

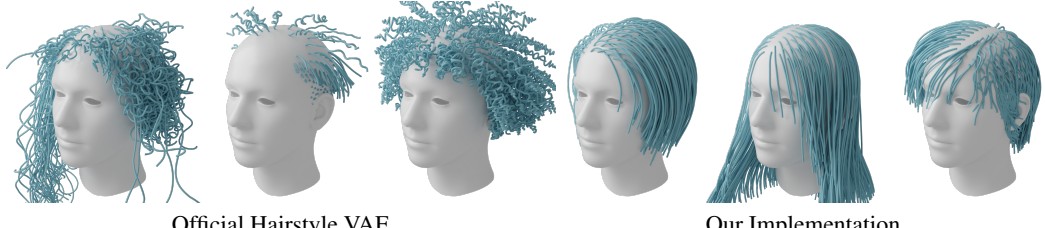

Official Hairstyle VAE                        Our Implementation

Figure 24: Comparison of our hairstyle VAE implementation with the official checkpoint of Groom-Gen (Zhou et al., 2023). Our implementation and the official version are sampled using the same Gaussian noise.

For the neural upsampler, as the authors cannot share their checkpoint, we can only use our own implementation as reference. However, as a GAN, we found this module very unstable to train, easily collapsing to weird outputs (see Fig. 7 and Fig. 27).

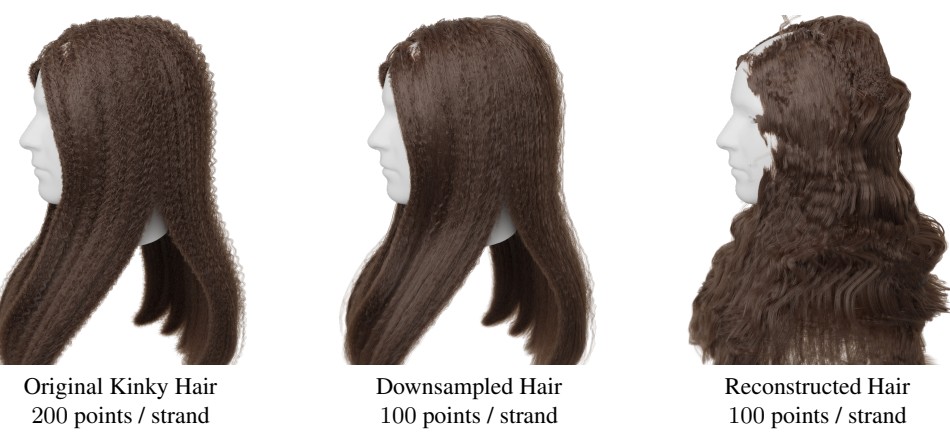

Original Kinky Hair          Downsampled Hair          Reconstructed Hair
200 points / strand          100 points / strand          100 points / strand

Figure 25: Verification of our GroomGen implementation on a manually made kinky hairstyle.

To assess whether our implementation of GroomGen can reproduce long kinky hairstyles, we manually created a test hairstyle (Fig. 25 1st column) which originally contains 200 points per strand to accurately capture the fine curls. We downsampled the strands to 100 points to fit our design, which led to a noticeable degradation in curl fidelity (Fig. 25 2nd column). We then searched for the optimal latent code to reconstruct the downsampled hair, yielding the results shown in the last column of Fig. 25. Despite these efforts, our implementation struggled to faithfully reconstruct the hairstyle with accurate fine details.

## D   ADDITIONAL RESULTS

### D.1   RANDOM HAIRSTYLE SYNTHESIS

In Fig. 26, we showcase several random guide strands generated by sampling the parameter space of $\vec{\theta}$ with Gaussian noise, highlighting that the results from StyleGAN2 exhibit greater diversity

compared to those generated by the PCA alternative discussed in our main paper. Given that PCA lacks constraints on the distribution of its subspace, obtaining reasonable guide strands by sampling its subspace with Gaussian noise is challenging, and our results indicate that most of them are collapsed into similar outputs.

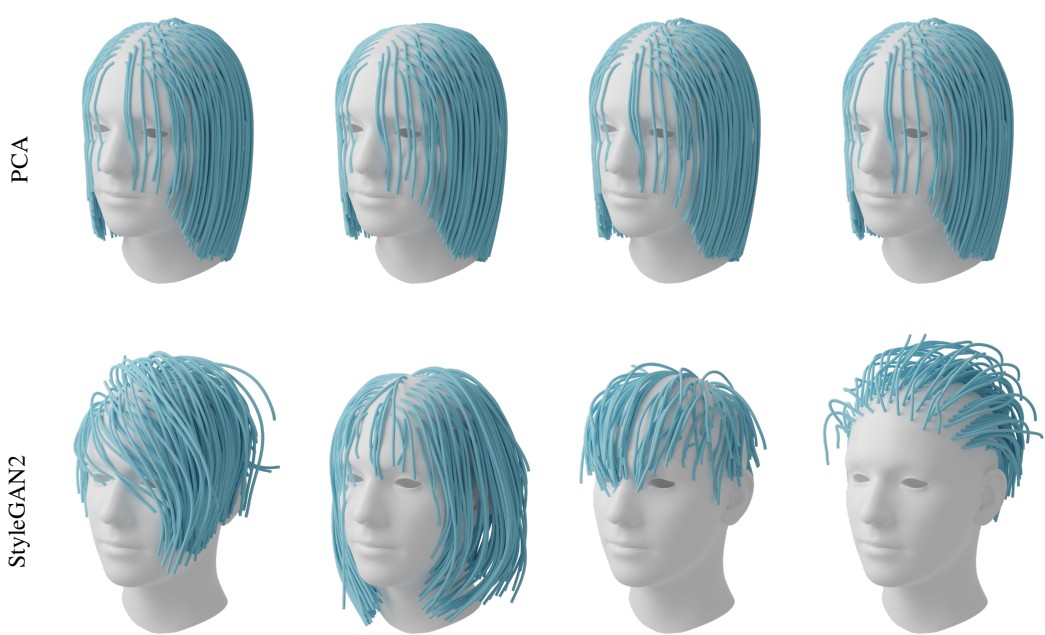

Figure 26: Guide strands synthesized from different Gaussian noise (Top: PCA; Bottom: Style-GAN2). Note that PCA samples are collapsed to similar results due to the distribution difference between Gaussian and PCA subspace.

In Fig. 27, we illustrate several random full hair models generated by sampling the parameter spaces of $\vec{\theta}$ and $\vec{\beta}$ with Gaussian noise, and compare these results to GroomGen (Zhou et al., 2023). Note that we sample our parameter space and GroomGen's latent space with the same Gaussian noise for a fair comparison. From our observation, GroomGen tends to generate hairstyles with implausible curls and flyaway strands, which do not appear in our results.

### D.2    HAIRSTYLE INTERPOLATION

In Fig. 28 we show hairstyle interpolation with different granularity. In the top row, we linearly interpolate the guide strand parameter $\vec{\theta}$, resulting in hairstyles with different global structures. Since the hair styling parameter $\vec{\beta}$ is kept fixed, local curl patterns, such as curls at the tips of strands, remain consistent. In the middle row, we fix the parameter $\vec{\theta}$ and linearly interpolate the parameter $\vec{\beta}$, thereby generating novel hairstyles ranging from straight to curly while maintaining a nearly identical haircut. In the bottom row, we jointly interpolate $\vec{\theta}$ and $\vec{\beta}$, demonstrating the transition from a short wavy hairstyle to a long straight hairstyle.

Additionally, we compare our method with (Weng et al., 2013) and (Zhou et al., 2018) on hairstyle interpolation. Since neither (Weng et al., 2013) nor (Zhou et al., 2018) are publicly available, we embed the 2 hairstyles selected by (Zhou et al., 2018) into our parameter space and jointly interpolate the projected $\vec{\theta}$ and $\vec{\beta}$ to obtain our results. Qualitative comparisons are provided in Fig. 29, demonstrating that our method achieves performance comparable to (Zhou et al., 2018), with both outperforming the results from (Weng et al., 2013). Note that in the middle column of interpolation, our method generates strands naturally covering the forehead, rather than severely intersecting with the head mesh like (Weng et al., 2013). Our starting and ending hairstyles are a bit different

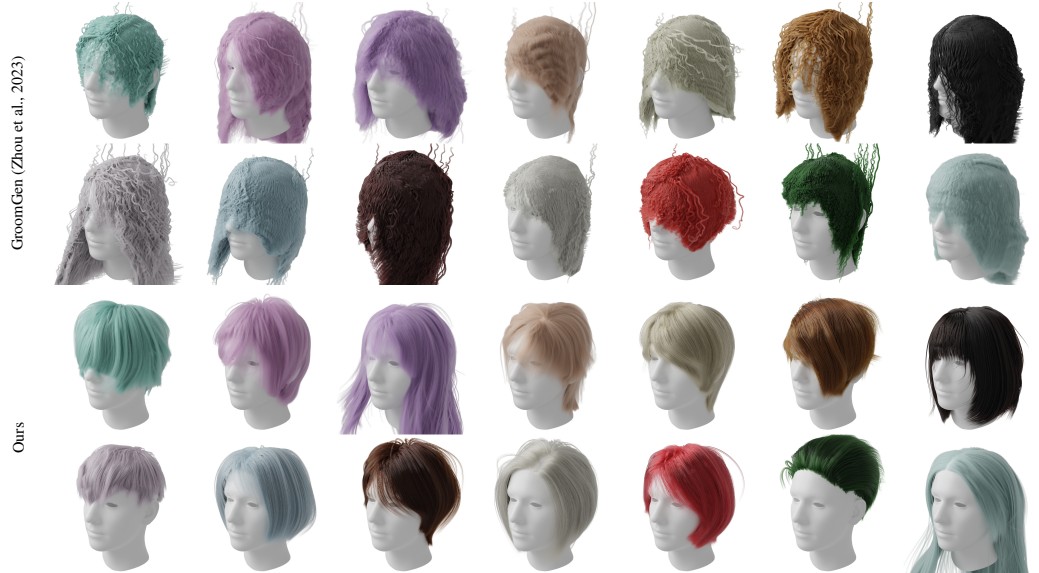

Figure 27: Full hair models synthesized from different Gaussian noise, with comparison to our implementation of GroomGen (Zhou et al., 2023) trained on the same dataset. Our model and GroomGen are sampled using the same Gaussian noise. Hair colors are manually assigned for aesthetic purposes.

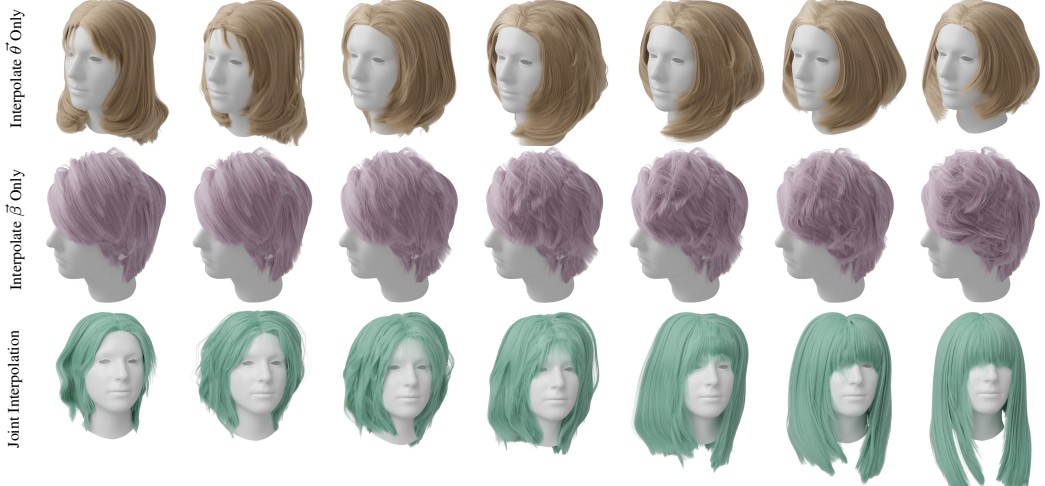

Figure 28: Hairstyle interpolation with different granularity. Top: interpolate $\vec{\theta}$ only and keep $\vec{\beta}$ fixed; Middle: interpolate $\vec{\beta}$ only and keep $\vec{\theta}$ fixed; Bottom: jointly interpolate $\vec{\theta}$ and $\vec{\beta}$.

from others, which is the result of our parameterization process, as some details cannot be fully reconstructed.

### D.3 SINGLE-VIEW HAIR RECONSTRUCTION

To test reconstruction for hair under dynamics, we run our algorithm on image sequences sampled from a video in (Yang et al., 2019), and compare our results to both (Yang et al., 2019) and HairStep (Zheng et al., 2023) in Fig. 30. Since Yang et al. (2019) have not released the training data or pre-trained model of their method, we can only use their provided videos for comparison.

Among these methods, our method achieves the best reconstruction quality regarding both the global hair structure and local curl patterns, while also avoiding artifacts such as bald areas observed in HairStep. Despite being trained solely on static data, our model demonstrates the ability to gener-

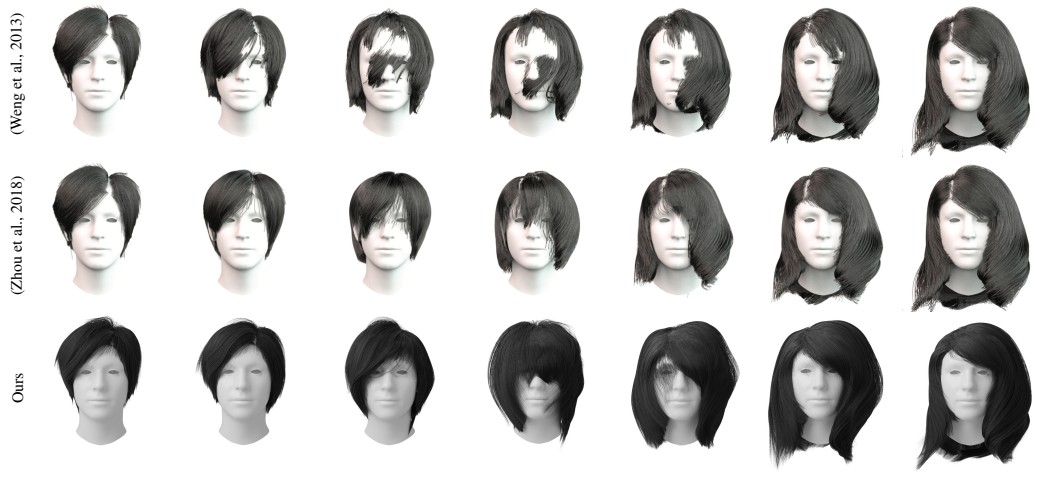

Figure 29: Interpolation comparison with (Weng et al., 2013) and (Zhou et al., 2018).

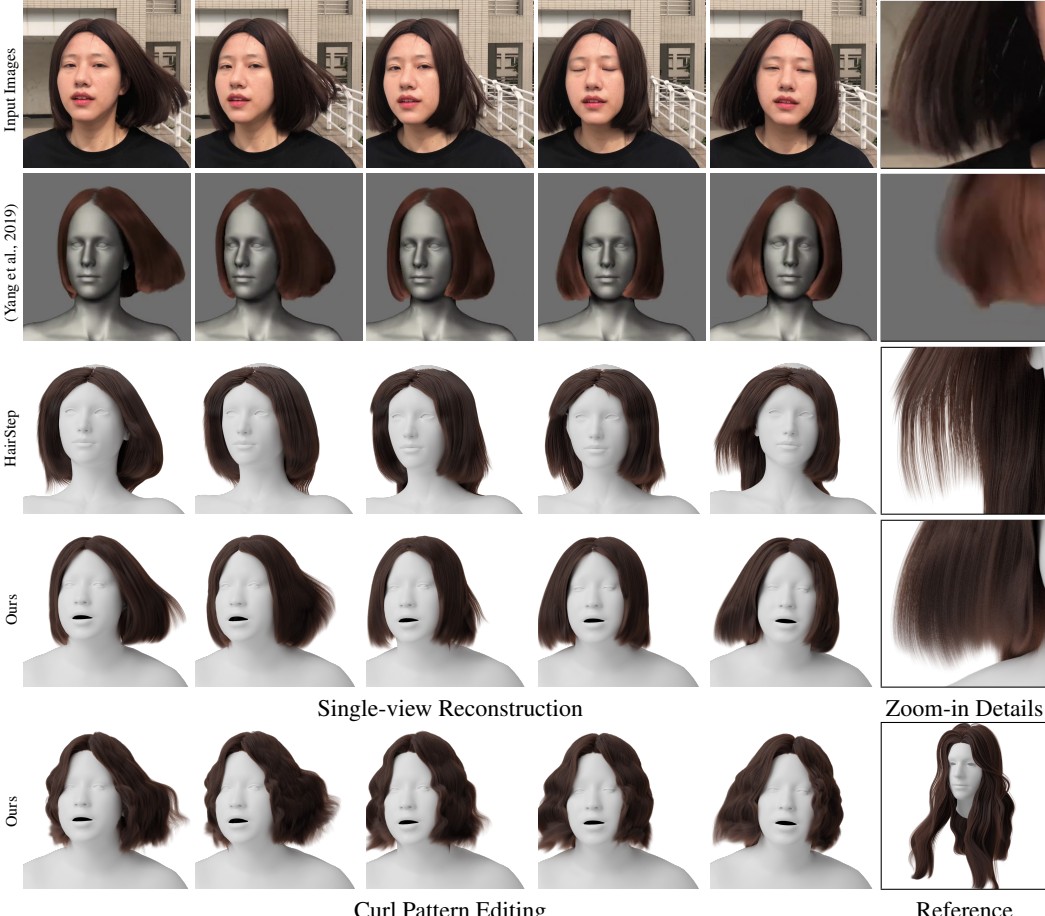

Figure 30: Single-view hair reconstruction and editing on image sequences, with comparison to (Yang et al., 2019) and HairStep (Zheng et al., 2023).

alize and capture the dynamic effects of hair in the images. We conjecture that it is because our disentangled guide strand parameters are flexible enough to capture large hair deformation variations. Additionally, since our reconstructed hairstyles are represented as parameters, we can substi-

tute their $\vec{\beta}$ parameters with that of a wavy hairstyle (shown as reference in the last row), thereby transferring the wavy patterns to the reconstructed results while preserving their overall structure.

Another recent work, HAAR (Sklyarova et al., 2023b), also demonstrates the capability to generate 3D hairstyles from input images. As a text-conditioned diffusion model, their pipeline involves first extracting hairstyle descriptions from the input image using LLaVA (Liu et al., 2024), then generates a strand-based hairstyle based on these descriptions. However, since textual descriptions cannot capture all the intricate details of a hairstyle, HAAR often fails to accurately reproduce the hairstyle from the input image, resulting in only a rough resemblance (see Fig. 31).

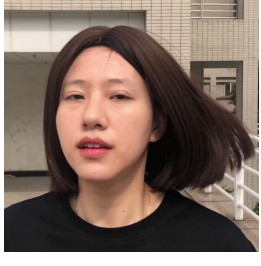 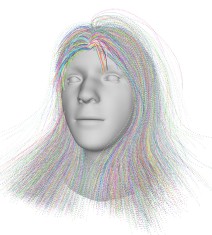

Input Image HAAR (Sklyarova et al., 2023b)

Figure 31: Single-view hair reconstruction of HAAR (Sklyarova et al., 2023b).

### D.4 HAIR-CONDITIONED IMAGE GENERATION

In Fig. 32 we present a comparison of image generation results with and without our input hair conditions. Note that we use the pre-visualization from MeshLab (Cignoni et al., 2008) as the structural image condition, which we found to perform better than the final renderings. Without our hair conditions, images generated with rough text prompts like *"wavy and short hair"* cannot guarantee the production of the desired hairstyle, and their hairstyles often vary with pose, resulting in a loss of multi-view consistency. Leveraging the rich information encoded in the 3D hair geometry, our hair conditions effectively address these issues, producing high-quality portrait images with more consistent hairstyles.

*"wavy and short hair, white sweater"*

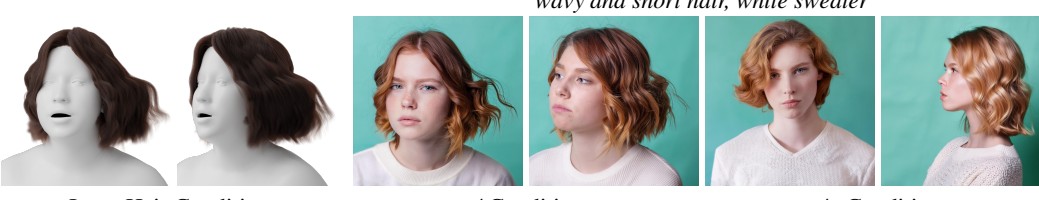

Input Hair Conditions w/ Conditions w/o Conditions

Figure 32: Comparison of image generation with and without hair conditions. These images are generated with the same text prompt *"wavy and short hair, white sweater"*. Additional text prompts like *"front face"* or *"side face"* are appended to assist the head pose in the generated images.

In Fig. 33 we further showcase some generated images conditioned on various input hairstyles. The texture colors in the input hairstyle renderings are only for aesthetic purposes and have no effect on the skin or hair colors in the generated images. Although structural conditioned image generation is not our focus and needs more future investigation on improving the structural alignment, this application reveals the potential of deploying current T2I models as a "neural renderer" (Tewari et al., 2020) within the traditional CG pipeline.

### E LIMITATIONS

Our work still suffers from several limitations. First, our training data is limited, as high-quality 3D hair assets are notoriously challenging to acquire. Consequently, certain intricate hairstyles, such as afro styles and various braided patterns, are underrepresented, which limits the generative capabilities of our model and leads to unnatural outputs (see Fig. 34). Addressing this issue requires the development of a systematic framework for efficiently capturing a diverse range of real-world data – a critical challenge that remains unresolved in the hair modeling area. However, PERM could serve as a pre-trained prior for efficient data capture, since it has the potential to fill in invisible

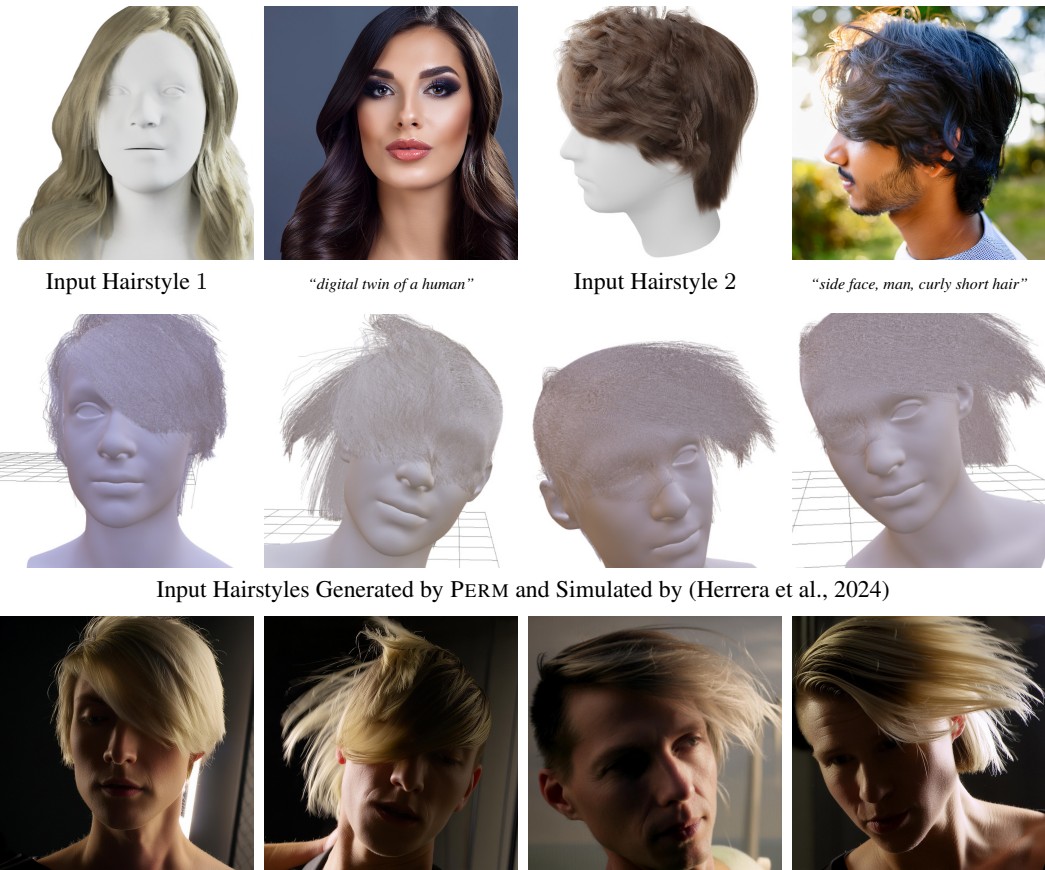

Input Hairstyle 1    *"digital twin of a human"*    Input Hairstyle 2    *"side face, man, curly short hair"*

Input Hairstyles Generated by PERM and Simulated by (Herrera et al., 2024)

*"a man with short hair is shaking his head"*

Figure 33: Hair-conditioned image generation using Adobe Firefly (Adobe, 2024).

parts of the hair, such as interiors and occluded regions, thereby accommodating more sparse image inputs. The captured hair data could also be used to fine-tune PERM, further bridging the domain gap between synthetic and real-world hairstyles.

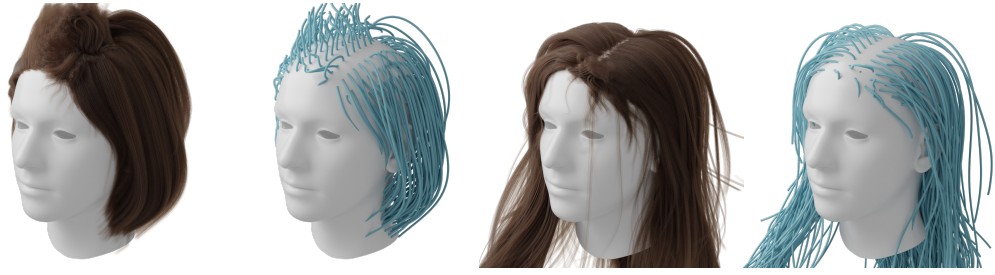

Figure 34: Unnatural hairstyles generated from random sampling.

Second, our current single-view hair reconstruction pipeline adopts a relatively straightforward approach, relying on an optimization process that primarily focuses on per-pixel strand information encoded in the 2D rendered images. These supervisions may fail to capture fine strand-level details, yielding suboptimal outputs in extreme cases (see Fig. 35). To enhance reconstruction quality, existing methods (Chai et al., 2012; Nam et al., 2019) could be incorporated to trace continuous strand segments from single-view or multi-view input images, providing a more reliable geometric clue for our optimization. Additionally, more advanced optimization techniques could be explored. For example, integrating patch-based losses, such as LPIPS (Zhang et al., 2018), may help encourage the

preservation of strand details at the patch level, thereby improving the fidelity of the reconstructed hairstyles.

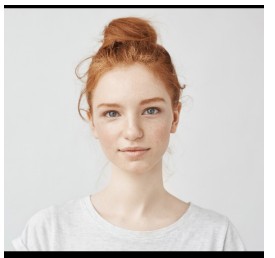 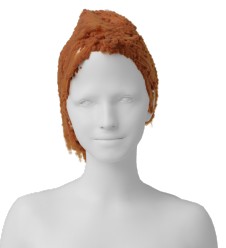 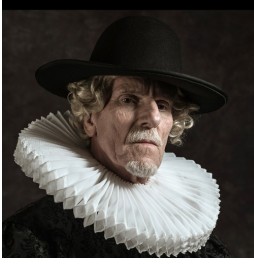 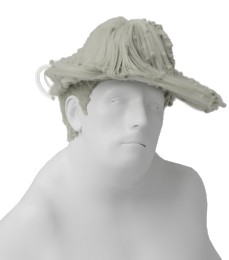

Figure 35: Suboptimal single-view reconstruction results in extreme cases, where the hair is tied in certain hairstyles such as buns (left) or heavily occluded by accessories such as hats (right).

