# OpenReview forum: "Perm: A Parametric Representation for Multi-Style 3D Hair Modeling"
_ICLR.cc/2025/Conference — ICLR 2025 Spotlight_

### Official Review · Reviewer_BeFM · 2024-10-29

**Soundness:** 3
**Presentation:** 3
**Contribution:** 3
**Rating:** 8
**Confidence:** 3

**Summary:**

The paper introduces **PERM**, a parametric representation for 3D human hair designed to improve a variety of hair-related applications. Unlike previous models that combine global hair structure and local curl patterns, PERM separates these elements using a **PCA-based strand representation** in the frequency domain. This allows for more precise control and editing by decomposing hair textures into different frequency levels, from low to high. Each frequency level is then parameterized with distinct generative models to mimic typical hair grooming stages. Experiments validate PERM's architecture, and the trained model is applied as a flexible prior for various tasks, including single-view hair reconstruction, hairstyle editing, and hair-conditioned image generation.

**Strengths:**

1. The proposed pipeline, as illustrated in Fig. 4, looks reasonable and straightforward.
2. The paper is well polished, and the experiments are sound.

**Weaknesses:**

N/A since I am not an expert in this area. A potential improvement direction, from my perspective, is to demonstrate the proposed representation on 3D generation tasks / other amortized pipelines, such as X-conditioned 3D avatar (with controllable hair) generation. E.g., as in Gaussian3Diff (ECCV 24'), the UV space can be used for 3D diffusion training for full head synthesis, and I think that pipeline is applicable to this proposed method.

**Questions:**

Regarding the overview pipeline, is the residual branch necessary, since the StyleGAN model with U-Net SR module can already generate pretty fine-grained results (intuitively). If this branch is removable, I believe the proposed pipeline will be more neat.

---

> ### Author Response · Authors · 2024-11-23
> **Response to Reviewer BeFM**
>
> Dear Reviewer BeFM,
>
> Thank you for your positive review and constructive feedback! Below, we address your concerns point by point:
>
> **Q:** *“A potential improvement direction, from my perspective, is to demonstrate the proposed representation on 3D generation tasks / other amortized pipelines, such as X-conditioned 3D avatar (with controllable hair) generation.”*
>
> **A:** Thank you for this insightful suggestion. While this direction is beyond the primary scope of our paper, we agree that our proposed representation could be integrated into 3D diffusion training based on our 2D uv texture design. With more 3D data available, it would certainly be feasible to train a 3D diffusion model for full-head synthesis with controllable hair modeling. We consider it as a promising research direction and look forward to seeing progress in this area by future research endeavors.
>
> ---
>
> **Q:** *“Regarding the overview pipeline, is the residual branch necessary, since the StyleGAN model with U-Net SR module can already generate pretty fine-grained results (intuitively).”*
>
> **A:** The residual branch is necessary because the output from our SR module lacks all high-frequency details, which needs to be complemented by the synthesized residual textures. In our revised paper, we include an example generated without the residual texture synthesis in **Figure 22**, which only recovers the global structure of the ground truth hairstyle. After adding the synthesized residual textures, the curl patterns are generated to better match the ground truth. Besides, our design with this residual branch also enables applications such as hairstyle interpolation with different granularity **(Figure 28)**. We can separately change the curliness of the hairstyle, thus generating novel hairstyles ranging from straight to curly while maintaining a nearly identical haircut.

---

> > ### Comment · Reviewer_BeFM · 2024-11-26
> > **Response to the author**
> >
> > Dear Authors,
> >
> > Thanks for your detailed clarification. Please include the above discussion in the related work / future work of revised version. I will keep my current rating and lean to accept this paper.

---

> > > ### Author Response · Authors · 2024-11-26
> > >
> > > Dear Reviewer BeFM:
> > >
> > > Thanks for your support of our work! We have uploaded a revised version of our manuscript, which includes these discussions in the Conclusion section.

---

> ### Author Response · Authors · 2024-11-25
> **A Gentle Reminder**
>
> Dear Reviewer BeFM:
>
> As the rebuttal phase approaches its end with only two days remaining for PDF revisions, we would like to kindly remind you of our responses to your comments and questions.
>
> We have carefully addressed all your feedback and have made thoughtful updates to the manuscript to address your concerns. We hope that our efforts meet your expectations and provide clarity on the points you raised.
>
> If you have any remaining questions or concerns, we would be happy to discuss them further and make additional revisions as needed.
>
> Thank you again for your time, thoughtful feedback, and invaluable contributions to improving our work.

---

### Official Review · Reviewer_SDZK · 2024-11-03

**Soundness:** 3
**Presentation:** 4
**Contribution:** 3
**Rating:** 8
**Confidence:** 3

**Summary:**

The paper proposes PERM, a lightweight and generic parametric representation of 3D human hair.  The proposed PERM method contributes to the field of 3D hair modeling by providing a lightweight, generic, and editable parametric representation that addresses the limitations of existing methods and enables versatile applications in various hair-related tasks.

Specifically, a novel parametric representation of 3D human hair, PERM, is introduced.   To independently control the global structure and curl pattern, a strand representation based on principal component analysis (PCA) in the frequency domain is proposed.  The learned principal components form an interpretable space for hair decomposition, where initial components capture low-frequency information, and subsequent components encode high-frequency details.  The paper follows previous literature to compute the UV of the scalp and map the geometry features of each strand onto a 2D texture based on its root position.  With the strand PCA representation, dominant coefficients fit and are stored in guide textures, and higher-order coefficients are stored in residual textures for detailed strand patterns.

**Strengths:**

The paper focuses on the hot topic of 3D hair parametric modeling and innovatively proposes a strand representation method based on Principal Component Analysis (PCA) in the frequency domain.  The paper is clearly articulated, with a rigorous technical approach and strong innovation.  The rendering results presented in the experimental stage effectively demonstrate the effectiveness of the method.

The authors have conducted extensive and in-depth experimental validations.  At the strand representation level, they have fully demonstrated the superiority of the PCA-based strand representation method compared to other representation approaches through comparative experiments.  In terms of full hair representation, the authors have compared the effectiveness with the Groom-Gen method.  Furthermore, the authors have showcased the broad application effects of the proposed parametric model in multiple tasks, including single-view hair reconstruction, perm parameter editing, and hair-conditioned image generation, fully proving the method's promising application prospects.

**Weaknesses:**

The paper only compared Groom-Gen and presented a limited number of samples, making it difficult to analyze the strengths and weaknesses of the two approaches comprehensively. To more accurately evaluate the methods, it is suggested that the number of samples in the comparison experiments be increased.

Furthermore, the paper did not discuss any failure cases of the proposed method. I am particularly interested in the failure cases of the method in random synthesis and single-view reconstruction tasks. Presenting these failure cases would help to elucidate the limitations of the proposed approach more clearly.

**Questions:**

As explained in the "weaknesses", my primary concerns lie in the experiments section.      I hope that the authors can provide detailed responses and explanations to the following specific issues:

* In the comparison with GroomGen, it is evident that all random sampling results generated by GroomGen exhibit markedly unrealistic strand structures.      How can this phenomenon be explained?

* In Figure 5, the rendering results of the PCA method and the authors' proposed Freq. PCA (Ours) methods are similar, to the extent that they are almost indistinguishable by visual inspection.      Could the authors further elaborate on the differences and merits of these two methods in terms of specific rendering details?

* What is the extent of the impact of Guide Texture Upsampling on the final rendering results?  Incorporating an ablation study on the upsampling module is crucial for comprehensively evaluating its significance and contribution.

* I suggest that the authors showcase failure cases encountered in random generated and single-view reconstruction tasks, and provide a thorough analysis and discussion of such cases.      This not only helps to uncover potential limitations of the current method but also offers valuable insights and references for future research endeavors.

---

> ### Author Response · Authors · 2024-11-23
> **Response to Reviewer SDZK**
>
> Dear Reviewer SDZK,
>
> Thank you for your detailed and thoughtful review! We sincerely appreciate your feedback and have carefully revised our paper to address your concerns. Below, we provide point-by-point responses to your comments:
>
> **Q:** *“The paper only compared Groom-Gen and presented a limited number of samples.”*
>
> **A:** Thanks for pointing it out! We choose GroomGen as it’s the most closely related work to ours (an unconditional generative model of 3D hair). To better illustrate the differences between GroomGen and our approach, we have included more random sample results in **Figure 27** of our revised paper *(14 samples for each method)*.
>
> ---
>
> **Q:** *“In the comparison with GroomGen, it is evident that all random sampling results generated by GroomGen exhibit markedly unrealistic strand structures. How can this phenomenon be explained?”*
>
> **A:** We have provided some possible reasons in **Appendix C.4**. To summarize:
> 1. While GroomGen’s strand VAE formulation with frequency features seems theoretically sound, its training strategy or loss design makes the trained model hard to fully represent the strand geometry. As demonstrated in **Figure 23**, both the official GroomGen strand VAE and our reimplementation failed to accurately reconstruct given strands.
> 2. GroomGen’s guide strands are quantitatively downsampled from the full hair, thus preserving all high-frequency details. This design choice increases the variation of guide strands, making the Hairstyle VAE more difficult to train. As shown in **Figure 24**, the trained Hairstyle VAE may produce some unnatural guide strands during random sampling.
> 3. GroomGen’s neural sampler uses a CNN-based GAN architecture, which we found very unstable during training. Consequently, the upsampled hairstyles often include unnatural curls and flyaway strands, as shown in **Figure 27**. In contrast, our method formulates this process as a deterministic mapping using U-Net, enabling stable supervised training and more natural upsampling results.
>
> ---
>
> **Q:** *“In Figure 5, the rendering results of the PCA method and the authors' proposed Freq. PCA (Ours) methods are similar, to the extent that they are almost indistinguishable by visual inspection. Could the authors further elaborate on the differences and merits of these two methods in terms of specific rendering details?”*
>
> **A:** Thanks for raising this important question! We choose Freq. PCA as our design because it is formulated in the frequency domain, which better aligns with the cyclical growth behavior of human hair. Our experiments also show that this approach preserves strand curliness more effectively than spatial-domain PCA, as reflected in the curvature error reported in **Table 1**. To further clarify this difference, we added a visual comparison in **Figure 15** in our revised paper, where strands are color-coded by their curvature error. The quantitative values are also provided, which are 2.671 and 2.341, respectively. These experiments further demonstrate that Freq. PCA better preserves strand curliness both qualitatively and quantitatively.
>
> ---
>
> **Q:** *“What is the extent of the impact of Guide Texture Upsampling on the final rendering results? Incorporating an ablation study on the upsampling module is crucial for comprehensively evaluating its significance and contribution.”*
>
> **A:** This experiment is already included in **Appendix C.2** and **Figure 21**, where we compare our super-resolution module with alternative interpolation methods. As shown in **Figure 21**, naive approaches such as nearest neighbor and bilinear interpolation consistently produce various artifacts, and our method achieves the most natural results that match the ground truth hair best.
>
> ---
>
> **Q:** *“I suggest that the authors showcase failure cases encountered in random generated and single-view reconstruction tasks, and provide a thorough analysis and discussion of such cases. This not only helps to uncover potential limitations of the current method but also offers valuable insights and references for future research endeavors.”*
>
> **A:** Thanks for this valuable suggestion! In the revised paper, we have added examples of typical failure cases in **Appendix E (Figure 34 and Figure 35)**, along with a detailed discussion of their causes and potential solutions. To summarize, these artifacts are primarily caused by limitations in our training data and the simplicity of our current optimization scheme. Expanding our training data with more realistic hairstyles and designing more advanced optimization techniques would likely improve both the generation and reconstruction quality of our model.

---

> > ### Comment · Reviewer_SDZK · 2024-11-26
> >
> > Thank the authors for the response. The supplemented experiments and discussions addressed most of my concerns.  Overall, I would maintain the original rating.

---

> > > ### Author Response · Authors · 2024-11-26
> > >
> > > Dear Reviewer SDZK:
> > >
> > > We greatly appreciate your recognition of our work's novelty and contribution. We sincerely thank you for your thoughtful feedback and are grateful for your positive recommendation!

---

> ### Author Response · Authors · 2024-11-25
> **A Gentle Reminder**
>
> Dear Reviewer SDZK:
>
> As the rebuttal phase approaches its end with only two days remaining for PDF revisions, we would like to kindly remind you of our responses to your comments and questions.
>
> We have carefully addressed all your feedback and have made thoughtful updates to the manuscript to address your concerns. We hope that our efforts meet your expectations and provide clarity on the points you raised.
>
> If you have any remaining questions or concerns, we would be happy to discuss them further and make additional revisions as needed.
>
> Thank you again for your time, thoughtful feedback, and invaluable contributions to improving our work.

---

### Official Review · Reviewer_vjoV · 2024-11-03

**Soundness:** 3
**Presentation:** 4
**Contribution:** 4
**Rating:** 8
**Confidence:** 3

**Summary:**

The paper proposed perm, a learning-based parametric representation for hair modeling. A PCA-based strand representation in the frequency domain is proposed to jointly model the global hair structure and local curl patterns, enabling hair editing and other downstream tasks. The authors have conductive comprehensive experiments, including comparisons to other representations, ablation studies, and demonstrations on downstream tasks, showing the great potential of the proposed hair parameterization model.

**Strengths:**

- The frequency domain PCA for hair strands is neat. Also, 10 coeffs for coarse strands and 54 coeffs for detailed curls is reasonable and straightforward.

- The proposed framework is meticulously designed. The model can be applied to differentiable optimization/rendering. I did not find an obvious fallback of the proposed framework.

- The evaluation of the paper is comprehensive, including comparisons with other methods, ablation studies on different key components, and exemplar downstream tasks. From my perspective, the authors successfully demonstrate the effectiveness/SOTA performance of the proposed representation.

**Weaknesses:**

I am not an expert in hair modeling and I did not find obvious fallbacks of the proposed framework. Below I listed the questions raised after reading the manuscript:

- From single view reconstruction results in Fig. 10, I find that the current SOTA is a good fit for the hair in the input image. However, the hair details are still misaligned. Is this restricted by model capacity given the current parameterization? (I understand hair modeling is very difficult). Are there any suggestions to improve the parameterization (e.g., modeling the random strands in the input image)?

- Is it possible (or how) to combine the proposed method with NeRF/3DGS for realistic full-head/full-body reconstruction?

**Questions:**

The two questions are listed above.

---

> ### Author Response · Authors · 2024-11-23
> **Response to Reviewer vjoV**
>
> Dear Reviewer vjoV,
>
> Thank you for your positive review and feedback! Below, we address your concerns point by point:
>
> **Q:** *“The hair details are still misaligned. Is this restricted by model capacity given the current parameterization?. Are there any suggestions to improve the parameterization?”*
>
> **A:** Thank you for raising this important question! In the revised paper, we have included a discussion of these limitations in **Appendix E**. Briefly, we hypothesize that the observed artifacts primarily arise from two key issues:
> 1. Our training data is limited, as high-quality 3D hair assets are notoriously challenging to acquire. This limited data distribution may restrict the generative capabilities of our model. Addressing this issue requires the development of a systematic framework to efficiently capture a diverse range of real-world data -- a critical challenge that remains unresolved in the hair modeling area.
> 2. Our current single-view hair reconstruction pipeline adopts a relatively straightforward approach, relying on an optimization process that primarily focuses on per-pixel strand information encoded in the 2D rendered images. These supervisions may fail to capture fine strand-level details, resulting in suboptimal output when the hair is tied in certain hairstyles such as buns or heavily occluded by accessories such as hats. To enhance reconstruction quality, better geometric clues and more advanced optimization techniques could be explored.
>
> Developing methods to model the random strands in input images is definitely another promising future research direction, as it increases the expressiveness of our parametric model. Achieving this, however, would require a clean separation of these random strands from the main hairstyle, as well as a suitable parameterization for them. Moreover, we would need some specific 3D training data, as most existing 3D hair models do not include such random flyaway strands. In conclusion, while this is an interesting direction to explore, it requires significant effort to make these solutions feasible.
>
> ---
>
> **Q:** *“Is it possible to combine the proposed method with NeRF/3DGS for realistic full-head/full-body reconstruction?”*
>
> **A:** Yes, this is another exciting direction for future work! As we discussed in the related work section, existing approaches such as *GaussianHaircut*, *GaussianHair*, *GroomCap*, and *HVH* have successfully combined strand-based hair with 3DGS/NeRF for 3D hair capture, where 3D Gaussians or neural primitives are attached to individual strands to model their appearance. In this context, our Perm model could serve as a pre-trained geometry prior, enabling the generation of plausible strands in invisible parts of a hairstyle (e.g., interior or occluded regions). This would provide a more robust geometric proxy for NeRF and 3DGS, thereby enhancing full-head and full-body reconstruction with a complete and controllable hair area.

---

> > ### Comment · Reviewer_vjoV · 2024-11-29
> >
> > Thanks the authors for their replies. I don't have further questions and I will keep my previous positive rating of 8.

---

> ### Author Response · Authors · 2024-11-25
> **A Gentle Reminder**
>
> Dear Reviewer vjoV:
>
> As the rebuttal phase approaches its end with only two days remaining for PDF revisions, we would like to kindly remind you of our responses to your comments and questions.
>
> We have carefully addressed all your feedback and have made thoughtful updates to the manuscript to address your concerns. We hope that our efforts meet your expectations and provide clarity on the points you raised.
>
> If you have any remaining questions or concerns, we would be happy to discuss them further and make additional revisions as needed.
>
> Thank you again for your time, thoughtful feedback, and invaluable contributions to improving our work.

---

> ### Author Response · Authors · 2024-11-27
> **Follow-up**
>
> Dear Reviewer vjoV:
>
> We sincerely appreciate the time and effort you have dedicated to reviewing our manuscript and for providing thoughtful and constructive feedback. With the deadline for revising the manuscript approaching on November 27th, we kindly ask if you have any remaining questions or require further clarification. We would be more than happy to provide additional information or address any remaining concerns.
>
> Thank you again for your thorough review and valuable insights.

---

> ### Author Response · Authors · 2024-11-29
>
> Dear Reviewer vjoV:
>
> Thanks for your support of our work! We sincerely appreciate your thoughtful feedback and are truly grateful for your positive recommendation!

---

### Official Review · Reviewer_4JLz · 2024-11-04

**Soundness:** 3
**Presentation:** 2
**Contribution:** 3
**Rating:** 6
**Confidence:** 4

**Summary:**

The paper presents PERM, a parametric model for representing 3D hair, designed to efficiently handle hair synthesis, editing, and reconstruction. PERM disentangles global hair structure and local curl patterns using a PCA-based strand representation in the frequency domain, enabling intuitive editing and precise control. The model architecture combines StyleGAN2 and VAE networks to generate guide and residual textures, capturing both broad and detailed features of hair. PERM performs well across tasks like single-view hair reconstruction and hairstyle editing, often surpassing task-specific methods. The approach is lightweight, computationally efficient, and achieves high fidelity, providing a robust framework for diverse hair modeling applications.

**Strengths:**

The strengths of the paper include its use of a PCA-based strand representation for 3D hair that efficiently disentangles global and local hair details, enabling precise control and intuitive editing. The PERM model’s architecture is computationally lightweight and achieves high fidelity with reduced memory and training requirements. It showcases versatility by performing well across various applications like single-view reconstruction and hairstyle editing, often matching or surpassing specialized, task-specific methods. Additionally, the framework’s modularity makes it adaptable for different hair modeling tasks, enhancing its practical value in digital human applications.

**Weaknesses:**

While the motivation of this paper is clear and the introduction is enjoyable to read, I am extremely frustrated with the forward-referencing (or no-referencing) writing style in Section 3. I often have to hold many questions until much later in the text, making it so easy to lose track of the mathematical flow. For example, Figure 4 starts with StyleGAN, but it is not properly introduced until Section 3.2. The same issue occurs with guide textures, residual textures, and many other components. The strand parameterization function S() , which is the last step in the diagram of Figure 4, is introduced first in the subsections, while preceding steps are explained in a recursive manner. The PCA coefficients, \gamma, are not explicitly present on the RHS of Eq 1. I understand that they are outputs of the guide textures, but I was so clueless when first looking at it until very late in Section 3. Additionally, the function  M()  on the LHS of Equation 1 was never properly introduced. I am concerned that this paper requires a major revision.

The paper presents the method as a parametric model, but unlike deterministic models such as SMPL or FLAME, it lacks a clear deterministic nature as it involves stochastic elements (e.g., generative processes or randomness in texture synthesis). This ambiguity raises questions about the consistency and reproducibility of the model’s outputs. You won’t be able to encode hair geometry into a fixed set of parameters and then decode it back accurately.

**Questions:**

Is  S = {p1 , p2 , . . . , pL }   the output of Equation 2? The way it is drafted in line 212 makes it seem like it should be an input to the subsequent equation.

Line 230 states that “Our analysis reveals that the first 10 PCA coefficients are enough to effectively capture the global structure of each strand, and the remaining 54 coefficients encode high-frequency details, such as curl patterns”. Where is this analysis? This is another place where I am frustrated by no-referencing.

---

> ### Author Response · Authors · 2024-11-23
> **Response to Reviewer 4JLz**
>
> Dear Reviewer 4JLz:
>
> Thank you for your detailed and thoughtful review! We truly appreciate your feedback and have carefully revised our paper to address your concerns. Below, we provide point-by-point responses to your comments:
>
> **Q:** *“While the motivation of this paper is clear and the introduction is enjoyable to read, I am extremely frustrated with the forward-referencing (or no-referencing) writing style in Section 3”*
>
> **A:** We apologize for the confusions caused by our writing style. We acknowledge that our previous writing is a bit misleading and unclear. In our revised paper, we have rewritten Section 3 to improve its clarity and flow. Specifically, we now introduce intuitive explanations for key terms (e.g., "guide textures," "residual textures," etc.) at the beginning of this section and include hyperlinks to the specific subsections where each concept is explained in detail. Please let us know if this version is clear enough to you. We keep "Strand Representation" as Section 3.1 because it introduces the concept of strand PCA coefficients, which is later used by all the textures.
>
> ---
>
> **Q:** *“The paper presents the method as a parametric model, but unlike deterministic models such as SMPL or FLAME, it lacks a clear deterministic nature as it involves stochastic elements (e.g., generative processes or randomness in texture synthesis).”*
>
> **A:** Sorry but we believe this is a misunderstanding. Unlike Diffusion Models, GANs and VAEs are **deterministic models**. This means that, given a fixed latent code, the model will always produce the same output once trained. The process of finding the latent code corresponding to a target image, often called *GAN Inversion*, is a well-studied problem, with a comprehensive survey in [1]. In our case, we have already performed such experiments in **Section 4.2** and **Figure 8**, where we encode hair geometry into a fixed set of Perm parameters through optimization. The results in **Figure 8** demonstrate a close match between the optimized outputs and the ground truth, validating the deterministic and expressive nature of our Perm model.
>
> ---
>
> **Q:** *“Is S = {p1 , p2 , . . . , pL } the output of Equation 2?”*
>
> **A:** Yes it is the output of Equation 2. Sorry for the confucions caused by our writing and we have updated the text to clarify this point – please see **Lines 216–217** in our revised paper.
>
> ---
>
> **Q:** *“Analysis of PCA coefficients”*
>
> **A:**  We apologize for the oversight in the previous manuscript. We have now included this analysis in **Appendix B.4** and **Figure 19**. In this analysis, we show a series of hairstyles reconstructed using different numbers of strand PCA coefficients. The results indicate that using 10 coefficients per strand strikes the best balance between smoothness and global structure preservation. This is why we chose to use 10 coefficients for the global structure and 54 coefficients for high-frequency details.
>
> ---
>
> > [1] Xia et al. "Gan inversion: A survey." IEEE transactions on pattern analysis and machine intelligence 45.3 (2022): 3121-3138.

---

> > ### Comment · Reviewer_4JLz · 2024-11-27
> >
> > Thank you for your responses. I believe the revision is clearer and better presented than the original manuscript.
> >
> > I hope the extended discussion period provides enough time to go over some details.
> >
> > While it is true that GANs have a deterministic generator, VAE decoders typically map a sampled latent to the data space by generating the means and variances of a multidimensional Gaussian, followed by a subsequent sampling process. In such cases, a fixed latent vector does not always correspond to the same output. Of course — it is possible to bypass sampling and use the mean of the Gaussian directly, in which case the decoder can be considered deterministic.
> >
> > I saw that VAEs were also used in the paper. Was this how the decoder was used in your pipeline?

---

> > > ### Author Response · Authors · 2024-11-27
> > >
> > > Dear Reviewer 4JLz:
> > >
> > > Yes, this reparameterization trick (generate mean and variance and then sample from them) is only used during training with KL loss to ensure that the latent space of the VAE (in our case, the space of $\vec{\beta}$) follows a normal distribution. However, after training, we only use the mean of the Gaussian as $\vec{\beta}$, ensuring that the process is deterministic.
> > >
> > > Moreover, because this process is both deterministic and differentiable, we further optimize $\vec{\beta}$ when fitting the Perm parameters to a given hairstyle. This optimization ensures that the fitted result better matches the input hairstyle. These implementation details are provided in **Appendix B.5**.
> > >
> > > We apologize for any confusion this may have caused and hope this explanation clarifies our approach. Please let us know if this answer is clear to you or not.

---

> > > > ### Comment · Reviewer_4JLz · 2024-11-28
> > > >
> > > > Thank you. I have raised my rating.

---

> > > > > ### Author Response · Authors · 2024-11-28
> > > > >
> > > > > Dear Reviewer 4JLz:
> > > > >
> > > > > We greatly appreciate your recognition of our work's novelty and contribution. We sincerely thank you for your thoughtful feedback and are grateful for your positive recommendation!

---

> ### Author Response · Authors · 2024-11-25
> **A Gentle Reminder**
>
> Dear Reviewer 4JLz:
>
> As the rebuttal phase approaches its end with only two days remaining for PDF revisions, we would like to kindly remind you of our responses to your comments and questions.
>
> We have carefully addressed all your feedback and have made thoughtful updates to the manuscript to address your concerns. We hope that our efforts meet your expectations and provide clarity on the points you raised.
>
> If you have any remaining questions or concerns, we would be happy to discuss them further and make additional revisions as needed. Otherwise, if you find our updates satisfactory, we kindly invite you to consider reevaluating your score.
>
> Thank you again for your time, thoughtful feedback, and invaluable contributions to improving our work.

---

### Author Response · Authors · 2024-11-23
**General Response to All Reviewers**

Dear Reviewers,

We thank you all for your valuable feedback! We have carefully considered your suggestions and revised our paper to address the concerns raised, where the updated sections are highlighted in red for clarity. In summary, our modifications include:
- **Improved clarity in Section 3:** We have rewritten Section 3 to improve readability. Intuitive explanations for each term have been added at the beginning of this section, and hyperlinks to specific subsections have been included for easier navigation. **(Reviewer 4JLz)**
- **PCA coefficients analysis:** An analysis of PCA coefficients has been added in **Appendix B.4 (Figure 19)** to validate our decision to use the first 10 PCA coefficients for global strand structure and the remaining 54 coefficients for high-frequency details. **(Reviewer 4JLz)**
- **Additional random sample results:** We have included more random sample results in **Figure 27** to better illustrate the differences between GroomGen and our approach. **(Reviewer SDZK)**
- **Comparison of PCA and Freq. PCA:** A visual comparison of PCA and Freq. PCA has been added in **Figure 15**, where strands are color-coded by their curvature error. Quantitative values are also provided, demonstrating that Freq. PCA better preserves strand curliness. **(Reviewer SDZK)**
- **Failure cases:** Typical failure cases encountered in random sampling and single-view hair reconstruction have been added to **Appendix E (Figure 34 and Figure 35)**, accompanied by a discussion of their causes and potential solutions. **(Reviewer vjoV, Reviewer SDZK)**
- **Residual branch validation:** An example generated without the residual texture synthesis branch has been added in **Figure 22**, demonstrating the necessity of this branch in our architecture design. **(Reviewer BeFM)**

Please let us know if you have any further questions or concerns about this revised version of our paper. Thank you again for your insightful feedback!

---

### Public Comment · ~Yuxiao_Zhou1 · 2025-04-10
**Comments and Clarifications from GroomGen**

Dear all,

I'm the first author of GroomGen, and I appreciate the opportunity to share a few clarifying comments regarding the comparisons with GroomGen presented in this paper.

In Section 4.2, under Comparison with GroomGen, the authors state that _"we also found that 100 points per strand is insufficient to represent kinky hairstyles, and GroomGen's output failed to capture the detailed curl patterns."_ I would like to respectfully clarify that the model and code we released to Perm (and made publicly available to all for research purposes) include an example script that generates three hair models - one of which closely resembles the hairstyle shown in Figure 25. I was surprised that this result was not demonstrated in the paper. Moreover, I believe the last column of Official Hairstyle VAE in Figure 24 also shows that GroomGen is capable of capturing a meaningful degree of curliness using 100 vertices per strand.

In Figure 23, a comparison is made between our official release and the authors' re-implementation. With all due respect, I find our official results to be perceptually more faithful to the ground truth, particularly in preserving curl structure. It's also important to note that our released model was trained on a __different dataset__ than the one used in this paper, making the comparison inherently unfair. The higher quantitative error observed in our results is therefore expected. Furthermore, as discussed in the original GroomGen paper, we argue that in the context of generative modeling, strand curvature is a more meaningful feature than precise vertex locations. This is the fundamental motivation behind __GroomGen's introduction of frequency-based strand shape encoding back in 2023__.

I fully understand how challenging it can be to reproduce results from complex systems, and I've made a genuine effort to provide accessible code and models. That said, it is disappointing to see GroomGen misrepresented while we have already released publicly available resources. As for the reimplementation of the densification step, I will refrain from commenting, as that component was not part of our release - and I fully respect Perm's effort to reproduce it.

I also noticed that several design choices in this paper - such as the use of frequency features (Section 3.1) and the prediction of guide weights for upsampling (Section 3.2.2) - bear a strong resemblance to techniques introduced in GroomGen. While it is entirely possible that these similarities are coincidental, I was nonetheless surprised that GroomGen was not cited in connection with these ideas.

I hope these clarifications contribute constructively to the discussion. Please feel free to point out if I have misunderstood anything.

Best regards,

Yuxiao

---

> ### Public Comment · ~Chengan_He1 · 2025-04-14
>
> Hi Yuxiao,
>
> Thank you for initiating this important discussion. We sincerely apologize if any technical aspects of GroomGen were overlooked, and we truly appreciate your engagement and the opportunity to clarify our contributions.
>
> **1. Regarding Our Design Choices and Novelty**
>
> Our core contribution lies in the **PCA-based strand representation** and the **interpretable disentanglement** of full hair generation into guide textures and residue textures, enabled by PCA parameterization. While we do incorporate frequency features to achieve an improvement in curvature preservation from an independent research finding, **we have never claimed the use of frequency features as our novel contribution**, out of respect for GroomGen. For clarity, our mention of GroomGen’s use of frequency features can be found in Section 4.1.
>
> Similarly, **we do not present the upsampling scheme as a novel contribution, nor do we consider it as a novel contribution of GroomGen**. Hair upsampling through neural interpolation is a widely adopted practice, first introduced in *“HairNet: Single-View Hair Reconstruction Using Convolutional Neural Networks” (Zhou et al., ECCV 2018)*, and subsequently used in works such as *“Real-Time Hair Simulation with Neural Interpolation” (Lyu et al., TVCG 2020)* and *“CT2Hair: High-Fidelity 3D Hair Modeling using Computed Tomography” (Shen et al., SIGGRAPH 2023)*. We reference GroomGen alongside other works in the first paragraph of Section 3.2.2 when discussing interpolation strategies.
>
> **2. Regarding Experimental Results and the GroomGen Implementation**
>
> Since you did not release the full model and dataset – only the frequency-based strand VAE and hairstyle VAE – we reimplemented the full pipeline and trained it on our publicly available dataset (https://github.com/c-he/GroomGen). If you identify any inaccuracies in our implementation, we sincerely welcome your feedback and are fully open to revising the code and re-running relevant experiments.
>
> Concerning **Figure 23**, which compares your released VAE and our replication, **we do not believe it is unfair as the testing hairstyle is from CT2Hair, a third-party dataset we did not use for training either**. Moreover, GroomHair contains more curly samples than USC-HairSalon, thus the result should favor the reconstruction of curly hairstyles as the one we tested. Regardless of dataset bias, **the reconstruction results from both the official GroomGen model and our reimplementation underperform compared to our PCA-based representation (see Figure 5), pointing to a consistent issue with the frequency-based strand VAE design**. Although we do acknowledge that using frequency features can help preserve strand curvature, and our experiment between PCA and Freq. PCA also supports this, experimentally we found the  frequency-based strand VAE in GroomGen cannot faithfully reconstruct hairstyles in our testing cases.
>
> If you still find unfairness, could you let us know which public dataset we should use to train our replications and which dataset we should use for testing the official version and our replicated version?
>
> Regarding **Figure 24**, our goal was to compare generated results from your released VAE and our replication using **the same Gaussian noise**. This is why we did not use your provided seed or test script. If you believe this comparison does not adequately reflect GroomGen’s generative capacity, we are happy to provide additional random sampling examples. We have also included our testing script for your review.
>
> **3. Regarding the Number of Points for Kinky Hair**
>
> We appreciate your correction regarding our earlier claim. After reviewing the kinky hairstyle you referenced, we agree that 100 points are sufficient to represent such styles within a certain length range. However, for longer kinky hairstyles, we find 100 points often insufficient for preserving high-frequency details and visual fidelity. **In our private dataset, styles of these complex types typically require over 200 points per strand**, and a model parameterized with 100 points cannot faithfully represent the hairstyle, even with frequency features. We apologize for any confusion and will revise the corresponding statement to clarify this point.
>
> In general, we greatly value your contributions to this field and are committed to constructive, transparent dialogue. If we have misunderstood any point or if further clarification is needed, please do not hesitate to let us know – we are happy to continue the discussion.

---

> > ### Public Comment · ~Chengan_He1 · 2025-04-14
> >
> > ```python
> > # generate new hair models
> >
> > for seed in range(100):
> >     hairstyle_latents = np.random.RandomState(seed).randn(1, 1, 1, 512).astype(np.float32)
> >     latent_maps, masks = hae_model.decode(hairstyle_latents)
> >     strands = sae_model.decode_latent_map_to_strands(latent_maps[0], masks[0], mask_threshold=0.9)
> >     strands = np.concatenate([strands, np.ones_like(strands[..., :1])], axis=-1)
> >     strands = (strands @ groomgen_to_pinscreen.transpose((0, 2, 1)))[..., :3]
> >     save_hair(f"outputs/seed{seed:04d}.abc", strands)
> > ```

---

> ### Public Comment · ~Yuxiao_Zhou1 · 2025-04-14
>
> Hi Chengan,
>
> Thank you for your response.
>
> I believe there may have been a slight misunderstanding. Please allow me to clarify: my intention is not to challenge the novelty or contributions of Perm (why do I?) - in fact, I am genuinely glad to learn that PCA can do well in your work.
>
> Rather, as wrote in my initial comment, I want to "share a few clarifying comments regarding the comparisons with GroomGen presented in this paper", because "it is disappointing to see GroomGen misrepresented while we have already released publicly available resources".
>
> To keep things organized, I'll follow-up in the same order as I asked, because the order reflects the objectivity of the questions:
>
> 1. Since you've acknowledged that Figure 25 was a misuse, I kindly ask you to remove that example and revise the corresponding claim in Section 4.2 and Section C.4. Using an image from our paper and asserting it as an impossible case is quite misleading. I mentioned Figure 24 for a similar reason: to show that 100 vertices can indeed represent curly hair models to a reasonable degree.
>
> 2. Regarding Figure 23, I'm afraid the main point might have been overlooked. My concern is that "(comparing with Perm's re-implementation) our official result is perceptually more faithful to the ground truth, particularly in preserving curl structure". In contrast, the result shown in the middle of Figure 23 appears noticeably broken, which suggests that GroomGen may not have been properly re-implemented. Perm states that the re-implementation was verified against the official checkpoints, and used Figure 25 "to assess whether our implementation of GroomGen can reproduce kinky hairstyles similar to those shown in the GroomGen teaser". Given the visual results in Figure 23 and the acknowledged misuse of Figure 25, I find it difficult to fully agree with that claim. It might be helpful to clearly differentiate between Perm's re-implementation and our original release in the paper to avoid potential confusion.
>
> 3. Lastly, when presenting non-trivial design choices, I always try to cite closely related work and briefly discuss the differences to clarify the motivations and novelty (if any) behind my approach. I understand that you may have a different writing philosophy, and of course, that is entirely your prerogative.
>
> Just to reiterate, my intention was never to challenge the novelty of your work, but rather to clarify GroomGen. Thank you again for the discussion, and I appreciate your time.
>
> Best regards,
> Yuxiao

---

> > ### Public Comment · ~Chengan_He1 · 2025-05-04
> >
> > Hi Yuxiao,
> >
> > Thank you for continuing this thoughtful discussion, and apologies for the delayed response due to recent ICLR travel. We have updated our paper to incorporate your feedback.
> >
> > Regarding our implementation of GroomGen, we have made our code publicly available for transparency and reproducibility. Based on our verification, we have found no evidence of fundamental issues with the implementation, especially for the strand VAE as shown in Figure 23. If you believe there are specific inaccuracies or technical errors, could you point them out directly? We genuinely welcome your feedback and are fully open to revisiting the code and re-running the relevant experiments. In response to your earlier comments, we have revised the manuscript to clearly indicate where comparisons are based on our own reimplementation. These updates are reflected in the captions of Figure 7, Figure 25, and Figure 27.
> >
> > For the strand point number, we have removed the example involving GroomGen and revised the corresponding claims in Section 4.2 and Appendix C.4. However, we would like to reiterate that for longer kinky styles, we found 100 points to be inadequate for faithfully capturing high-frequency curl details. In our private dataset, these complex hairstyles typically require 200 or more points per strand to ensure visual fidelity.
> >
> > Finally, regarding the design choice in Perm: as previously clarified, we do not claim novelty for the elements you referenced, and we have cited the relevant prior work where appropriate. While interpretations of writing style and emphasis may vary, we have made every effort to accurately acknowledge existing literature and contributions and wanted to make this point transparent and clear in the response.
> >
> > Please feel free to share any thoughts on the updated version or let us know if you have further questions or suggestions. We remain open to continuing this dialogue and value your engagement.

---

### Meta-Review · Area_Chair_1N3M · 2024-12-20

**Metareview:**

The paper proposes a parametric representation for 3D hair using a PCA-based strand representation in the frequency domain. This approach enables precise control and editing by decomposing hair textures into different frequency levels. The proposed representation facilitates various applications, including hair synthesis, editing, and reconstruction.

The main strength of the paper lies in its novelty. While the reviewers raised several issues requiring improvement or clarification, there is no significant weakness. These concerns were effectively addressed in the rebuttal.

In summary, the paper makes a significant contribution by proposing a novel representation of 3D hair and demonstrating its effectiveness through extensive experiments.

**Additional Comments On Reviewer Discussion:**

The paper mainly received positive feedback in the initial reviews, with some issues raised.

One reviewer highlighted issues with the presentation. These were addressed in the revision, which significantly improved the clarity and structure of the paper.

Concerns were raised about the consistency and reproducibility of the model’s outputs due to potential stochastic elements. The rebuttal resolved this by detailing adjustments to make the proposed method deterministic.

Reviewers suggested adding more comparisons with GroomGen. The rebuttal included additional comparisons and provided further discussions on the limitations of GroomGen.

As recommended by a reviewer, the revision included examples of failure cases and a discussion of potential improvements.

A reviewer questioned the necessity of the residual branch. The rebuttal explained its importance and supported the explanation with an example, effectively addressing the concern.

The rebuttal effectively addressed most of these concerns, and after the discussion stage, all reviewers expressed positive opinions about the paper.

---

### Decision · Program_Chairs · 2025-01-22

Accept (Spotlight)